# Statistical Efficiency of Thompson Sampling for Combinatorial Semi-Bandits

**Pierre Perrault**
Inria Lille — ENS Paris-Saclay
pierre.perrault@inria.fr

**Etienne Boursier**
ENS Paris-Saclay
etienne.boursier1@gmail.com

**Vianney Perchet**
ENSAE — Criteo AI Lab
vianney.perchet@normalesup.org

**Michal Valko**
DeepMind Paris — Inria Lille
valkom@deepmind.com

## Abstract

We investigate stochastic combinatorial multi-armed bandit with semi-bandit feedback (CMAB). In CMAB, the question of the existence of an efficient policy with an optimal asymptotic regret (up to a factor poly-logarithmic with the action size) is still open for many families of distributions, including mutually independent outcomes, and more generally the multivariate *sub-Gaussian* family. We propose to answer the above question for these two families by analyzing variants of the *Combinatorial Thompson Sampling* policy (CTS). For mutually independent outcomes in $[0, 1]$, we propose a tight analysis of CTS using Beta priors. We then look at the more general setting of multivariate sub-Gaussian outcomes and propose a tight analysis of CTS using Gaussian priors. This last result gives us an alternative to the *Efficient Sampling for Combinatorial Bandit* policy (ESCB), which, although optimal, is not computationally efficient.

## 1 Introduction

Stochastic multi-armed bandits (MAB) Robbins [1952], Berry and Fristedt [1985], Lai and Robbins [1985] are decision-making frameworks in which a learning *agent* acts sequentially in an uncertain environment. At every round $t \in \mathbb{N}^*$, the agent must select one arm from a pool of $n$ arms, denoted by $[n] \triangleq \{1, \dots, n\}$, using a learning *policy* based on the feedback collected from the previous rounds. Then it obtains as feedback a reward (also called *outcome*) $X_{i,t} \in \mathbb{R}$ — a random variable sampled from $\mathbb{P}_{X_i}$, independently from previous rounds — where $i$ is the selected arm and $\mathbb{P}_{X_i}$ is a probability distribution — unknown to the agent — of mean $\mu_i^*$. The goal for the agent is to maximize the cumulative reward over a total of $T$ rounds ($T$ may be unknown[1]). The performance metric of a policy is the regret, i.e., the expectation of the difference over $T$ rounds of the cumulative reward between the policy that always picked the arm with the highest expected reward and the learning policy. MAB models the classical dilemma between exploration and exploitation, i.e., whether to continue exploring arms to obtain more information (and thus strengthen the confidence in the estimates of the distributions $\mathbb{P}_{X_i}$), or to use the information gathered by playing the best arm according to the observations so far.

In this paper, we study stochastic combinatorial multi-armed bandit (CMAB) [Cesa-Bianchi and Lugosi, 2012], which is an extension of MAB where the agent selects a *super arm* (or *action*) $A_t \in \mathcal{A} \subset \mathcal{P}([n])$ at each round $t$. The set $\mathcal{A}$ is the *action space*, defined as a collection of subsets

of the (base) arms. The kind of reward and feedback varies depending on the problem at hand. We consider the *semi-bandit* setting, where the feedback includes the outcomes of all base arms in the played super arm. Formally, the agent observes[2] $\mathbf{X}_t \odot \mathbf{e}_{A_t} \triangleq (X_{i,t}\mathbb{I}\{i \in A_t\})_{i \in [n]}$ and the reward, given the choice of $A_t$, is a function of $\boldsymbol{\mu}^* \odot \mathbf{e}_{A_t}$ (traditionally, the reward is linear and equal to $\mathbf{e}_{A_t}^\top \boldsymbol{\mu}^*$, but our analysis goes beyond this setting). In recent years, CMAB has attracted a lot of interest (see e.g. Gai et al. [2012], Chen et al. [2013, 2016], Kveton et al. [2015], Wang and Chen [2017], Perrault et al. [2019b, 2020a]), particularly due to its wide applications in network routing, online advertising, recommender system, influence marketing, etc.

In CMAB, the whole joint distribution of the vector of outcomes $\mathbf{X}$ matters, contrary to standard MAB where only the marginals are sufficient to characterize a problem instance. For example, the following two extreme problem instances are distinct within the CMAB framework:

($i$) Each $\mathbb{P}_{X_i}$ is sub-Gaussian and the arm distributions are mutually independent, i.e., $\mathbb{P}_{\mathbf{X}} = \otimes_{i \in [n]} \mathbb{P}_{X_i}$.

($ii$) Each $\mathbb{P}_{X_i}$ is sub-Gaussian but the stochastic dependencies between the arm distributions are "worst case": the performance metric is the supremum of the regret over all possible dependencies between the marginals.

Those two settings are indeed different as two different lower bounds on the asymptotic (in $T$) regret can be derived. In particular, the regret scales as $\Omega(n \log(T)/\Delta)$ for the setting ($i$), and as $\Omega(mn \log(T)/\Delta)$ for ($ii$), where $\Delta$ is the minimum gap in the expected reward between an optimal super arm and any non-optimal super arm, and where $m \triangleq \max_{A \in \mathcal{A}} |A|$.

Many CMAB policies are based on the *Upper Confidence Bound* (UCB) approach, extending the classical UCB policy [Auer et al., 2002] from MAB to CMAB. This type of approach uses an optimistic estimate $\boldsymbol{\mu}_t$ of $\boldsymbol{\mu}^*$ (i.e., for which the reward function is overestimated), lying in a well-chosen confidence region. For setting ($ii$), there exist UCB-style policies that match the lower bound mentioned above. An example of such policy is *Combinatorial Upper Confidence Bound* (CUCB) [Chen et al., 2013, Kveton et al., 2015], that uses a Cartesian product of the individual confidence intervals of each arm as a confidence region. For setting ($i$), Combes et al. [2015] provided the UCB-style policy *Efficient Sampling for Combinatorial Bandit* (ESCB), that uses the assumption of mutual independence between arm distributions in order to build a tighter ellipsoidal confidence region around the empirical mean, which helps to better restrict the exploration. Degenne and Perchet [2016b] gave the following generalization of setting ($i$):

($iii$) The joint probability $\mathbb{P}_{\mathbf{X}}$ is $\mathbf{C}$-sub-Gaussian, for a positive semi-definite matrix $\mathbf{C} \succeq 0$, i.e., $\mathbb{E}\left[e^{\boldsymbol{\lambda}^\top (\mathbf{X} - \boldsymbol{\mu}^*)}\right] \leq e^{\boldsymbol{\lambda}^\top \mathbf{C} \boldsymbol{\lambda}/2}$, for all $\boldsymbol{\lambda} \in \mathbb{R}^n$.

In this case, they provided the policy OLS-UCB, leveraging this additional assumption and such that it essentially reduces to ESCB in the specific case of diagonal matrix $\mathbf{C}$ with a regret bound of $\mathcal{O}\left(\log^2(m)n \log(T)/\Delta)\right)$ (so it matches the above lower bound up to a polylogarithmic factor in $m$). We refer the reader to Table 1 for an overview of the above regret (lower) bounds.

In some CMAB problems, the action space $\mathcal{A}$ and the reward function are simple enough for the existence of an exact *oracle* that takes as input a vector $\boldsymbol{\mu} \in \mathbb{R}^n$ and outputs the solution of the combinatorial problem (associated to the mean vector $\boldsymbol{\mu}$), with a polynomial time complexity $\mathcal{O}(\text{poly}(n))$. Under this assumption (referred to as Assumption 1), CUCB, that plays the action $A_t = \text{Oracle}(\boldsymbol{\mu}_t)$ at round $t$, is efficient to implement, and has a $\mathcal{O}(\text{poly}(n))$ time complexity per round. In that case, the setting ($ii$) is therefore essentially solved. On the other hand, this is not true for the settings ($i$) and ($iii$), as ESCB needs to solve a difficult combinatorial problem in each round (NP-Hard in general [Atamtürk and Gómez, 2017]).

The inefficiency of ESCB triggered some attempts to implement an efficient version: Perrault et al. [2019a] proposed an efficient approximation method for implementing ESCB in the case the action space has a *matroid* structure: they prove a time complexity of $\mathcal{O}(\text{poly}(n))$ while keeping the same

Table 1: Factor in front of $n \log(T)/\Delta$ in the regret bound ($\mathcal{O}(\cdot)$ for upper bounds), computationally inefficient policies are printed with a subscript $*$, setting $(iii)$ is for $\mathbf{C}$ diagonal, CLIP CTS-GAUSSIAN is for linear reward functions, and with only $\boldsymbol{\lambda} \in \mathbb{R}_+^n$ in $(iii)$. Our results are printed in bold, see Theorem 1, Theorem 2, Theorem 3 related to CTS-BETA, CTS-GAUSSIAN, CLIP CTS-GAUSSIAN respectively.

|        | CUCB | ESCB$_*$ | CTS-BETA | CTS-GAUSSIAN | CLIP CTS-GAUSSIAN | Lower bound |
|--------|------|----------|----------|--------------|-------------------|-------------|
| $(i)$   | $m$ | $\log^2(m)$ | $\mathbf{\log^2(m)}$ | $\mathbf{\log^2(m)}$ | $\mathbf{\log^2(m)}$ | $\Omega(1)$ |
| $(ii)$  | $m$ | $m$ | - | $\mathbf{\log^2(m)m}$ | $\mathbf{m}$ | $\Omega(m)$ |
| $(iii)$ | $m$ | $\log^2(m)$ | - | $\mathbf{\log^2(m)}$ | $\mathbf{\log^2(m)}$ | $\Omega(1)$ |

regret rate. However, this improvement is mitigated by the fact that CUCB reaches the optimal regret rate $\mathcal{O}(n \log(T)/\Delta)$ for the special case of matroid semi-bandits [Anantharam et al., 1987, Kveton et al., 2014, Talebi and Proutiere, 2016]. Recently, Cuvelier et al. [2020] provided another approach for approximating ESCB for a wide variety of action spaces, including the matching bandit setting [Gai et al., 2010] and the online shortest path problem [Liu and Zhao, 2012], where CUCB is not known to be better than ESCB. However, their policies are still computationally expensive when $T$ is large, since the time complexity at round $t$ is of order $\mathcal{O}(t \cdot \text{poly}(n))$.

Another line of research is to find an efficient alternative to ESCB. One of the most promising candidate is *Thompson Sampling* (TS). Although introduced much earlier by Thompson [1933], the theoretical analysis of TS for frequentist MAB is quite recent: Kaufmann et al. [2012], Agrawal and Goyal [2012] gave a regret bound matching the UCB policy theoretically. Moreover, TS often performs better than UCB in practice, making TS an attractive policy for further investigations. For CMAB, TS extends to *Combinatorial Thompson Sampling* (CTS). In CTS, the unknown mean $\boldsymbol{\mu}^*$ is associated with a belief (a prior distribution) updated to a posterior with the Bayes'rule, each time a feedback is received. In order to choose an action at round $t$, CTS draws a sample $\boldsymbol{\theta}_t$ from the current belief, and plays the action given by $\text{Oracle}(\boldsymbol{\theta}_t)$. CTS is attractive also because its time complexity is $\mathcal{O}(\text{poly}(n))$ under Assumption 1. Recently, for the setting $(i)$ with bounded outcomes, Wang and Chen [2018] proposed an analysis of CTS-BETA, which is CTS where the prior distribution is chosen to be a product of $n$ Beta distributions. They proved two regret upper bounds depending on the class of reward functions:

$$\mathcal{O}\left(\frac{n\sqrt{m}\log(T)}{\Delta}\right) \text{ in the linear case and } \mathcal{O}\left(\frac{nm\log(T)}{\Delta}\right) \text{ in the general case.} \qquad (1)$$

Although the aforementioned upper bound in the linear reward case outperforms the one of CUCB, it doesn't match the one of ESCB. To summarize, and despite many efforts, the existence of a policy that is both optimal (up to a polylogarithmic factor in $m$) and efficient in the setting $(i)$ or $(iii)$ is still an open problem, which we tackle in this paper.

**Further related work** We refer the reader to Wang and Chen [2018] for further related work on TS for combinatorial bandits, and particularly for Gopalan et al. [2014], that provided a frequentist high-probability regret bounds for TS with a general action space and a general feedback model — Komiyama et al. [2015], that investigated TS for the $m$-sets action space — Wen et al. [2015], that studied TS for contextual CMAB problems, using the Bayesian regret metric (see also Russo and Van Roy [2016]).

## 1.1 Contributions

We first improve the result of Wang and Chen [2018] by providing the regret upper bound $\mathcal{O}\left(\log^2(m)n \log(T)/\Delta\right)$ for CTS-BETA in the setting $(i)$ with bounded outcomes. This bound is valid even for non linear reward functions. Our main contribution is a regret bound for the setting $(iii)$. We propose an efficient policy called CTS-GAUSSIAN, that is CTS where the prior distribution is chosen to be a multivariate Gaussian. An analysis of CTS-GAUSSIAN allows us to obtain a regret bound reducing to $\mathcal{O}\left(\log^2(m)n \log(T)/\Delta\right)$ for a diagonal sub-Gaussian matrix. When the reward function is linear, we generalize the setting $(iii)$ assuming only $\boldsymbol{\lambda} \in \mathbb{R}_+^n$. This allows us to get rid of negative correlations between the outcomes (as in Perrault et al. [2020b]), and focus on positive correlations. We propose in this setting the policy CLIP CTS-GAUSSIAN, where the score is truncated from below with the empirical mean, and from above with the UCB. Truncations from above are not

---

**Algorithm 1** CTS-BETA

---
    **Initialization**: For each arm $i$, let $a_i = b_i = 1$.
    **For all** $t \geq 1$:
        Draw $\boldsymbol{\theta}_t \sim \otimes_{i \in [n]} \mathrm{Beta}(a_i, b_i)$, and play $A_t = \mathrm{Oracle}(\boldsymbol{\theta}_t)$.
        Get the observation $\mathbf{X}_t \odot \mathbf{e}_{A_t}$, and draw $\mathbf{Y}_t \sim \otimes_{i \in A_t} \mathrm{Bernoulli}(X_{i,t})$.
        For all $i \in A_t$ update $a_i \leftarrow a_i + Y_{i,t}$ and $b_i \leftarrow b_i + 1 - Y_{i,t}$.

---

necessary, but can limit optimism, especially when positive correlations are significant. We obtain an improved regret bound for CLIP CTS-GAUSSIAN, where negative correlations no longer appear in the regret bound and where, in setting $(ii)$, the extra $\log^2(m)$ factor present in the regret bound of CTS-GAUSSIAN disappears. All these results are summarized and compared to other state-of-the-art policies in Table 1.

## 2  Model

CMAB is formally introduced as follows. Consider a random process $(\mathbf{X}_t) \overset{iid}{\sim} \mathbb{P}_{\mathbf{X}}$, where $\mathbb{P}_{\mathbf{X}}$ is a distribution — unknown to the agent — of random vectors in $\mathbb{R}^n$, with unknown mean $\boldsymbol{\mu}^*$. At each round $t \in [T]$, the agent chooses a super arm (or action) $A_t \in \mathcal{A} \subset \mathcal{P}([n])$ based on the history of observations $\mathcal{H}_t \triangleq \sigma(\mathbf{X}_1 \odot \mathbf{e}_{A_1}, \ldots, \mathbf{X}_{t-1} \odot \mathbf{e}_{A_{t-1}})$ and a possible extra source of randomness (we denote by $\mathcal{F}_t$ the filtration containing $\mathcal{H}_t$ and the extra randomness of round $t$ — in particular, $A_t \in \mathcal{F}_t$). The feedback received is then $\mathbf{X}_t \odot \mathbf{e}_{A_t}$ and the associated expected reward of the agent at that stage is $r(A_t, \boldsymbol{\mu}^*)$, for some known function $r$. The objective of the agent is to minimize the regret, defined for a policy $\pi$ as

$$\forall T \geq 1, \quad R_T(\pi) \triangleq \mathbb{E}\left[\sum_{t=1}^{T} \Delta_t\right],$$

where $\Delta_t \triangleq \Delta(A_t) \triangleq r(A^*, \boldsymbol{\mu}^*) - r(A_t, \boldsymbol{\mu}^*)$ with $A^* \in \arg\max_{A' \in \mathcal{A}} r(A', \boldsymbol{\mu}^*)$. As stated in the introduction, we will assume the following:

**Assumption 1.** *The agent has access to an oracle with a time complexity $\mathcal{O}(\mathrm{poly}(n))$ such that for any mean vector $\boldsymbol{\mu}$, $\mathrm{Oracle}(\boldsymbol{\mu}) \in \arg\max_{A \in \mathcal{A}} r(A, \boldsymbol{\mu})$.*

Similar to Chen et al. [2016], we assume that the function $r$ satisfies the following smoothness property.

**Assumption 2.** *There exists a constant $B$, such that for every super arm $A \in \mathcal{A}$ and every pair of mean vectors $\boldsymbol{\mu}$ and $\boldsymbol{\mu}'$, $|r(A, \boldsymbol{\mu}) - r(A, \boldsymbol{\mu}')| \leq B\|\mathbf{e}_A \odot (\boldsymbol{\mu} - \boldsymbol{\mu}')\|_1$.*

For an arm $i \in [n]$, we define the number of time $i$ has been chosen at the beginning of round $t$ as $N_{i,t-1} \triangleq \sum_{t' \in [t-1]} \mathbb{I}\{i \in A_{t'}\}$. We also define the following quantities, that will be useful in the expression of an upper bound on the regret:

    $m^* \triangleq \min_{A \in \arg\max_{A' \in \mathcal{A}} \mathbf{e}_{A'}^\mathsf{T} \boldsymbol{\mu}^*} |A|$ is the minimum size of an optimal action,
    $\Delta_{i,\min} \triangleq \min_{A \in \mathcal{A}, \, \Delta(A) > 0, \, i \in A} \Delta(A)$, is the minimal gap of an action containing $i \in [n]$,
    $\Delta_{\min} \triangleq \min_{i \in [n]} \Delta_{i,\min}$, is the minimal arm-gap and
    $\Delta_{\max} \triangleq \max_{A \in \mathcal{A}} \Delta(A)$ is the maximal gap.

## 3  Regret bound for CTS-BETA in setting $(i)$

In this section, we consider the following assumption on top of the CMAB setting from section 2.

**Assumption 3.** *The outcomes $X_i$ are bounded (in $[0, 1]$, w.l.o.g.), and are mutually independent (we are thus in a special case of $(i)$).*

For this problem, we consider CTS-BETA in Algorithm 1, which is described as follows. The prior is set to be a product of $n$ beta distributions (being thus uniform over $[0, 1]$ initially). Notice, this prior is conjugate to a product of Bernoulli distributions. After the agent get an observation $X_{i,t}$,

---

**Algorithm 2** CTS-GAUSSIAN

---

**Input**: The vector $\mathbf{D}$, and a parameter $\beta > 1$.

**Initialization**: Play each arm once (if the agent knows that $\boldsymbol{\mu}^* \in [a,b]^n$, this might be skipped)

**For every subsequent round** $t$:

    Draw $\boldsymbol{\theta}_t \sim \otimes_{i\in[n]} \mathcal{N}\big(\overline{\mu}_{i,t-1}, N_{i,t-1}^{-1}\beta D_i\big)$ $(\theta_{i,t} \sim \mathcal{U}[a,b]$ if $N_{i,t-1}=0)$.

    Play $A_t = \text{Oracle}(\boldsymbol{\theta}_t)$.

    Get the observation $\mathbf{X}_t \odot \mathbf{e}_{A_t}$, and update $\overline{\boldsymbol{\mu}}_{t-1}$ and counters accordingly.

---

it first binarizes it by sampling $Y_{i,t} \sim \text{Bernoulli}(X_{i,t})$ (the regret of the problem defined by the observations $Y_{i,t}$ is the same because $\mathbb{E}[Y_{i,t}] = \mu_i^*$). Then the prior is updated using Bayes' rule with each sample $Y_{i,t}$. When choosing a super arm at round $t$, the agent draws $\boldsymbol{\theta}_t$ from the beta belief, and then plugged it into the oracle, which outputs the super arm $A_t$ to play.

The main result of this section is Theorem 1, that improves the regret bound of Wang and Chen [2018] for CTS-BETA.

**Theorem 1.** *The policy $\pi$ described in Algorithm 1 has regret $R_T(\pi)$ of order*

$$\mathcal{O}\left( \sum_{i\in[n]} \frac{B^2 \log^2(m)\log(T)}{\Delta_{i,\min}} \right).$$

The proof of Theorem 1, as well as the complete non-asymptotic upper-bound is postponed to Appendix A. Our analysis incorporates two novelties that we detail in the two following paragraphs.

**An improved leading term**    (cf. Step 3 of the proof of Theorem 1 in Appendix A) We define the *empirical average* of each arm $i \in [n]$ at the beginning of round $t$ as $\overline{\mu}_{i,t-1} \triangleq \sum_{t'\in[t-1]} \frac{\mathbb{I}\{i\in A_{t'}\}Y_{i,t'}}{N_{i,t-1}}$. Notice that this empirical average definition differs from the one that is classically used in CMAB, since samples $Y_{i,t'}$ are used rather than $X_{i,t'}$. The improved dependence in $m$ in the leading term of Theorem 1 (compared to (1)) is a consequence of two ingredients. The first is the following concentration inequality (see Appendix A, Lemma 2), which improves that of Wang and Chen [2018] by extending it to the case of non-linear reward. Indeed, we rather control the $\ell_1$ norm in this case, instead of the $\ell_\infty$-norm, which leads to a tighter bound.

$$\mathbb{P}\left[ \left\| \mathbf{e}_{A_t} \odot \big(\boldsymbol{\theta}_t - \overline{\boldsymbol{\mu}}_{t-1}\big) \right\|_1 \geq \sqrt{\frac{1}{2}\log(|\mathcal{A}|2^m T)\sum_{i\in A_t}\frac{1}{N_{i,t-1}}} \,\bigg|\, \mathcal{H}_t \right] \leq 1/T. \tag{2}$$

The second ingredient is a more careful handling of the square-root term in the above probability, based on a method similar to the one in Degenne and Perchet [2016b].

$T$**-independent term**    (cf. Step 4 of the proof of Theorem 1 in Appendix A) Similarly to Wang and Chen [2018], our regret bound also contains an exponential term that is constant in $T$. Note, however that the term of Wang and Chen [2018] is of order $\mathcal{O}(\varepsilon^{-2m^*-2})$, whereas ours is of order $\mathcal{O}\big(\varepsilon^{-4m^*-2}\big)$, where $\varepsilon \in (0,1)$ is of order $\Delta_{\min}/(m^*)^2$. This discrepancy is due to the correction of a minor negligence inaccuracy in their Lemma 7, where they assume, at the end of the proof, that one could decorrelate the counters from the outcomes received. We manage to circumvent this issue by doing a careful union bound over the counters. It is this union bound that brings a larger dependence in this constant term. An additional discussion is deferred to the end of Appendix A.

## 4   Regret bound for CTS-GAUSIAN in setting $(iii)$

In this section, we consider the setting from section 2, with a more general sub-Gaussian family for $\mathbf{X} \in \mathbb{R}^n$. More precisely, we make the following similar assumption as in Degenne and Perchet [2016b]. Proposition 1 gives two examples included in this assumption (see Appendix B for a proof).

**Assumption 4.** *There exists a vector $\mathbf{D} \triangleq (D_1, \dots, D_n) \in \mathbb{R}_+^n$ known to the agent such that*

$$\forall A \in \mathcal{A},\ \forall \boldsymbol{\lambda} \in \mathbb{R}^n\ s.t.\ \boldsymbol{\lambda} = \boldsymbol{\lambda} \odot \mathbf{e}_A,\quad \mathbb{E}\Big[e^{\boldsymbol{\lambda}^\top(\mathbf{X}-\boldsymbol{\mu}^*)}\Big] \leq e^{\boldsymbol{\lambda}^\top\mathbf{D}\odot\boldsymbol{\lambda}/2}.$$

**Motivation for sub-Gaussian outcomes** In the same way as boundedness generalizes to sub-Gaussianity in 1d, we have that if $\mathbf{X}$ is a.s. in a compact $\mathcal{K}$, it is $\mathbf{C}$-sub-Gaussian, with $\mathbf{C}$ built from the John's ellipsoid of $\mathcal{K}$. In this case, $D_i$ is computed with a linear maximization over $\mathcal{A}$. In particular, $\mathcal{K} = B_{\ell_\infty}(0,1)$ gives $D_i = m$, and $\mathcal{K} = B_{\ell_2}(0,1)$ gives $D_i = 1$. We can also use other structures on the outcomes to have $D_i$, such as negative dependence (as we will see in our shortest path experiments, in section 5).

**Proposition 1.** *Assumption 4 encompasses the $\kappa_i^2$-sub Gaussian outcomes with worst case dependencies between the arm distributions, taking $D_i = \kappa_i^2 m$. It also captures $\mathbf{C}$-sub-Gaussian outcomes with a known sub-Gaussian matrix $\mathbf{C}$ (setting $(iii)$), taking $D_i = \max_{A \in \mathcal{A}, \, i \in A} \sum_{j \in A} |C_{ij}|$.*

For the above setting, we provide CTS-GAUSSIAN in Algorithm 2, where we define the empirical mean of arm $i$ at round $t \geq 1$ as $\overline{\mu}_{i,t-1} \triangleq \sum_{t' \in [t-1]} \frac{\mathbb{I}\{i \in A_{t'}\} X_{i,t'}}{N_{i,t-1}}$. This algorithm is comparable to Algorithm 1 but considers a Gaussian prior for each arm. Notice, the Gaussian family is *self-conjugate*, so except in the Gaussian-outcomes case, we do not rely on exact conjugated prior here. Although this is not surprising — since it is known that TS can work without exact conjugate prior with respect to the outcomes — obtaining an upper bound on the regret of the policy CTS-GAUSSIAN is non-trivial and constitutes our main contribution. We state our main result in Theorem 2.

**Theorem 2.** *The policy $\pi$ described in Algorithm 2 has regret $R_T(\pi)$ of order*

$$\mathcal{O}\left( \sum_{i \in [n]} \frac{B^2 D_i \log^2(m) \log(T)}{\Delta_{i,\min}} \right).$$

The proof of Theorem 2, as well as the complete non-asymptotic upper-bound is postponed to Appendix C. Nonetheless, in the following paragraphs, we provide some insights and highlight the novelty of our analysis.

**Main proof challenges** In the setting of the previous section, the outcomes are independent in $[0,1]$ and an important step in Algorithm 1 was to transform the outcomes into binary variables in order to be consistent with the posterior. Here, outcomes are no longer independent. In addition to that, we cannot transform the outcomes into Gaussian variables in the same way as in Algorithm 1. These two points are the main technical challenges to address in our analysis.

**Stochastic dominance** Before providing details on how we deal with the above challenges, first recall that the standard analysis (in the case of a factorized prior, that we have here[3]) consists in bounding the expected number of rounds needed for the sample $\boldsymbol{\theta}_t$ to be close to the true mean $\boldsymbol{\mu}^*$ on a certain set $Z \subset A^*$, i.e., for the event $\{\|(\boldsymbol{\mu}^* - \boldsymbol{\theta}_t) \odot \mathbf{e}_Z\|_\infty > \varepsilon\}$ to happen. We let $\mathfrak{T}_t(Z)$ denote the complementary event. As for the proof of Theorem 1, we can condition on the history to rewrite this expected number of rounds and then upper bound it as

$$\mathbb{E}\left[ \sum_{t \geq 1} (t-1) \mathbb{P}[\neg \mathfrak{T}_t(Z) | \mathcal{H}_t] \prod_{j=1}^{t-1} \mathbb{P}[\mathfrak{T}_j(Z) | \mathcal{H}_j] \right]$$

$$\leq \mathbb{E}\left[ \sup_{t \geq 1} \frac{1}{\mathbb{P}[\neg \mathfrak{T}_t(Z) | \mathcal{H}_t]} \right] - 1 \leq \sum_{Z' \subset Z, \, Z' \neq \emptyset} \mathbb{E}\left[ \sup_{t \geq 1} \prod_{i \in Z'} \left( \frac{1}{\mathbb{P}[|\theta_{i,t} - \mu_i^*| \leq \varepsilon | \mathcal{H}_t]} - 1 \right) \right].$$

Now, using the fact that the conditional distribution of $\theta_{i,t} - \overline{\mu}_{i,t-1}$ is symmetric and depends only on the counter $N_{i,t-1}$, we obtain that the probability $\mathbb{P}[|\theta_{i,t} - \mu_i^*| \leq \varepsilon | \mathcal{H}_t]$ is a monotonic function of the deviation $|\overline{\mu}_{i,t-1} - \mu_i^*|$. Let us emphasize that this property of the Gaussian prior used is crucial and that it is not obvious to transfer the same technique to a beta prior. To sum up, we have to control a term of the form $\mathbb{E}\left[\sup_{t \geq 1} \prod_{i \in Z'} g_i(|\overline{\mu}_{i,t-1} - \mu_i^*|)\right]$, where $g_i$ are non-negative increasing functions. Our approach is to prove that $(|\overline{\mu}_{i,t-1} - \mu_i^*|)_i$ is *weakly stochastically dominated* by $\left( \sqrt{\frac{\beta D_i}{N_{i,t-1}}} |\eta_i| \right)_i$, where $\boldsymbol{\eta} \sim \otimes_i \mathcal{N}(0,1)$, which is the same vector but where the empirical mean is

built with independent Gaussian outcomes instead. Notice, independence is crucial to be able to factorize the expectation $\mathbb{E}\big[\prod_{i \in Z'} g_i\big]$, in the same way as in the proof of Theorem 1. We recall two equivalent definitions of $\mathbf{U}$ is weakly stochastically dominated by $\mathbf{V}$, see Shaked and Shanthikumar [2007] for more details and properties of dominances,

- For all non-negative, non-increasing functions $f_i$, it holds $\mathbb{E}[\prod_i f_i(U_i)] \leq \mathbb{E}[\prod_i f_i(V_i)]$.
- For any vector $\mathbf{x}$, it holds $\mathbb{P}[\mathbf{U} \geq \mathbf{x}] \leq \mathbb{P}[\mathbf{V} \geq \mathbf{x}]$.

The first point applied to $g_i$'s (and up to the supremum over $t$) is a simple way to obtain the aforementioned wanted control. Thus, it's enough to prove the second point, which is a consequence of the sub-Gaussianity of outcomes given by Assumption 4 and some concentration inequality. Finally, we circumvent the supremum over $t \geq 1$ issue thanks to Doob's optional sampling theorem for non-negative super-martingales (see Durrett [2019], Theorem 5.7.6).

**Importance of using a factorized prior in our analysis** Note that in Algorithm 2, the samples $\theta_{i,t}$ are independent, while the outcomes are not necessarily independent. This independence is in fact crucial in order to be able to start the analysis in the same way as in the proof of Theorem 1 (recall that Algorithm 1 also uses a factorized prior). More precisely, a factorized prior allows us to link the filtered regret against the event $\mathfrak{S}_t(Z) \wedge \mathfrak{T}_t(Z)$ to the expected number of rounds needed for $\neg\mathfrak{T}_t(Z)$ to occur (see (3) in Step 4 of the proof of Theorem 1 in Appendix A for a definition of $\mathfrak{S}_t(Z)$). Indeed, without the factorized prior, the two events $\mathfrak{S}_t(Z), \mathfrak{T}_t(Z)$ would no longer be independent conditionally to the history, and the term $1/\mathbb{P}[\neg\mathfrak{T}_t(Z)|\mathcal{H}_t]$ obtained in the previous paragraph would then be replaced by $1/\mathbb{P}[\neg\mathfrak{T}_t(Z)|\mathfrak{S}_t(Z), \mathcal{H}_t]$, which is much more difficult to deal with. To the best of our knowledge, it is unknown how to get the desired bound when $\mathfrak{S}_t(Z)$ and $\mathfrak{T}_t(Z)$ are not independent conditionally to the history.

## 4.1 CLIP CTS-GAUSSIAN for the linear reward case

In this subsection, we make the following assumptions on top of Section 2.

**Assumption 5.** *The reward function is linear, defined as* $r(A, \boldsymbol{\mu}) \triangleq \mathbf{e}_A^\intercal \boldsymbol{\mu}$.

**Assumption 6.** *The agent knows a matrix* $\boldsymbol{\Gamma} \succeq 0$ *s.t.* $\forall \boldsymbol{\lambda} \in \mathbb{R}_+^n$, $\mathbb{E}\big[e^{\boldsymbol{\lambda}(\mathbf{X}-\boldsymbol{\mu}^*)}\big] \leq e^{\boldsymbol{\lambda}^\intercal \boldsymbol{\Gamma} \boldsymbol{\lambda}/2}$.

Notice that Assumption 6 slightly generalises the setting from Degenne and Perchet [2016b]. Requiring $\boldsymbol{\lambda} \in \mathbb{R}_+^n$ allows us to take $D_i = \max_{A \in \mathcal{A},\ i \in A} \sum_{j \in A}(0 \vee \Gamma_{ij})$, so that negative correlations are no longer harmful. $D_i$ can still be too large (and thus $\boldsymbol{\theta}_t$ might be over-sampled), so we cap $\boldsymbol{\theta}_t$ with the score $\boldsymbol{\mu}_t$ used by CUCB. The resulting policy is CLIP CTS-GAUSSIAN, where the score $\boldsymbol{\theta}_t$ is replaced by $\overline{\boldsymbol{\mu}}_{t-1} \vee \boldsymbol{\theta}_t \wedge \boldsymbol{\mu}_t$ before we plug it into Oracle, where $\mu_{i,t} = \overline{\mu}_{i,t-1} + \sqrt{\Gamma_{ii}\frac{2(\log(t)+4\log\log(t))}{N_{i,t-1}}}$. CLIP CTS-GAUSSIAN enjoys the following regret bound.

**Theorem 3.** *The policy* CLIP CTS-GAUSSIAN *has regret of order*
$$\mathcal{O}\left(\sum_{i \in [n]} \frac{\big(D_i \log^2(m) \wedge m\Gamma_{ii}\big)\log(T)}{\Delta_{i,\min}}\right).$$

Not only $D_i$ is improved through the above relaxation, but also, the leading term is never worse than the one of CUCB. The proof and the complete non-asymptotic upper-bound is delayed to Appendix D. We note that we rely heavily on reward linearity to analyse this clip version, not only using monotony to restrict the controls to the $\mathbb{R}_+^n$ directions (and thus to cap from bellow the sample by the empirical mean), but also using the oracle's invariance property $\text{Oracle}(\boldsymbol{\mu}) = \text{Oracle}\big(\boldsymbol{\mu} + \boldsymbol{\delta} \odot \mathbf{e}_{\text{Oracle}(\boldsymbol{\mu})}\big)$, with $\boldsymbol{\delta} \geq 0$, to cap the sample from above by the UCB.

**Comparison with the OLS-UCB analysis of Degenne and Perchet [2016b]** The leading term in the regret bound given from Theorem 3 is comparable to the one for OLS-UCB from Degenne and Perchet [2016b]. Indeed, we recall that they obtained a factor of order $\Gamma_{ii}\big((1-\gamma)\log^2(m) + \gamma m\big)$, with $\gamma \triangleq \max_{A \in \mathcal{A}} \max_{(i,j) \in A^2, i \neq j}(0 \vee \Gamma_{ij})/\sqrt{\Gamma_{ii}\Gamma_{jj}}$, where we have $\big(D_i \log^2(m) \wedge m\Gamma_{ii}\big)$. When $\gamma \in \{0, 1\}$ (this is the case when we are in the settings (i) and (ii) respectively), these two terms coincide. When $\gamma \in (0, 1)$, they are incomparable in general. We can still see that our variance term $D_i$ is always lower than their $\Gamma_{ii}((1-\gamma) + \gamma m)$, i.e., that our bound rate is lower than $\log^2(m)$ times theirs.

# 5 Experiments

Before describing the experiments carried out, notice that in the CTS-GAUSSIAN policies, $\beta > 1$ is an artefact of the analysis and can in practice be taken equal to $1$. This is what we did in our experiments.

**The shortest path problem**  We compare our CTS policies to CUCB and CUCB-KL, for the shortest path problem on the road chesapeake network [Rossi and Ahmed, 2015]. This network contains 39 nodes and $n = 170$ edges. $\mathcal{A}$ is the set of paths from an origin to a destination in the network. We choose a linear reward, so that an efficient Oracle exists for this problem. We choose $\boldsymbol{\mu}^*$ uniformly in $[-1,0]^n$ and then normalize its sum so that $\sum_i \mu_i^* = -s$, where $s$ is unknown to the agent. The parameter $s$ stands for the global network traffic (e.g., the total number of vehicles in the network). We run two experiments, one with $-\mathbf{X} \sim \otimes_i \mathrm{Bernoulli}(-\mu_i^*)$ and another with $-\mathbf{X} \sim \otimes_i \mathrm{Bernoulli}(-\mu_i^*)$ conditionally on $\sum_i X_i = -s$. They are presented in Figure 1. Since the outcomes are not mutually independent in this last experiment, we use (CLIP) CTS-GAUSSIAN rather than CTS-BETA, where we take $D_i = 1/4$, using that for any $\boldsymbol{\lambda} \in \mathbb{R}_+^n$, $\mathbb{E}\left[e^{\boldsymbol{\lambda}^\mathsf{T}\mathbf{X}}\right] \leq \prod_{i \in [n]} \mathbb{E}\left[e^{\lambda_i X_i}\right]$ (see e.g., Borcea et al. [2009], corollary 4.18). It is clear from the experiments that CTS policies outperform both CUCB and CUCB-KL. In the second experiment, we see that CLIP CTS-GAUSSIAN and CTS-GAUSSIAN are very similar — which is not surprising because $D_i$ is not large here (unlike in the next experiment) — and that for a small $s$, CUCB-KL becomes competitive, since the kl is much larger than the quadratic divergence in that case.

**Comparison to ESCB for the matching problem**  We consider here a comparison between (CLIP) CTS-GAUSSIAN, CUCB and ESCB (we refer the reader to Wang and Chen [2018] for a comparison between CTS-BETA and ESCB). Since ESCB is computationally intractable, we limit ourselves to a toy matching problem on the complete bipartite graphs $K_{4,4}$, with $\mathbf{X} \sim \mathcal{N}(\boldsymbol{\mu}^*, (c\mathbb{I}\{i \neq j\}+\mathbb{I}\{i = j\})_{ij})$, where this covariance is known to the agent. Our results are shown in Figure 2, where we observe that CLIP CTS-GAUSSIAN (resp. ESCB) is slightly better for $c$ small (resp. large), thus reaching the best of both worlds. This is because a large $c$ forces CLIP CTS-GAUSSIAN to oversample (as evidenced by CTS-GAUSSIAN whose performance is even worse than CUCB for $c = 1$). We also recorded the computation time for larger instances (see Table 2), and observe the efficiency of CUCB and CLIP CTS-GAUSSIAN compared to ESCB.

**Correlated vs independent prior in practice**  We briefly discussed the use of a correlated prior in footnote 3, with covariance $\left(C_{ij} N_{ij,t-1}^{-1} N_{i,t-1}^{-1} N_{j,t-1}^{-1}\right)_{ij}$, mentioning that the policy would perform better than using an independent prior. We ran additional empirical comparisons to assess this, plotting the results in Figure 3 where we also compared with a common prior policy approach [Agrawal et al., 2017], i.e., with covariance $\left(N_{i,t-1}^{-1/2} N_{j,t-1}^{-1/2}\right)_{ij}$.[4] As expected, the correlated prior policy is better than the independent one (when outcomes are correlated). This motivates the theoretical study of such policy for future work. The common prior approach is comparable to the correlated prior one on the matching problem, but it is outperformed in the worst-case scenario of a separate action space $\mathcal{A} = \left\{\{km + 1, \ldots, (k+1)m\} \mid k \in \left\{0, \ldots, \frac{n}{m} - 1\right\}\right\}$ with independent outcomes. This is because such problem reduces to a classical MAB problem with a covariance scaled up by a factor $m$, whereas the common prior approach has a variance scaled up by a factor $m^2$.

Table 2: Computation time per round (ms), with $c = 0.3$, $T = 100$, averaged over 5 simulations.

|  | $K_{3,3}$ | $K_{4,4}$ | $K_{5,5}$ | $K_{6,6}$ | $K_{7,7}$ | $K_{8,8}$ |
|---|---|---|---|---|---|---|
| CUCB | 0.39 | 0.64 | 1.23 | 1.65 | 2.45 | 3.88 |
| CLIP CTS-GAUSSIAN | 0.50 | 0.80 | 1.75 | 1.79 | 3.30 | 5.42 |
| ESCB | 0.45 | 1.93 | 10.3 | 75.6 | 541 | 4694 |

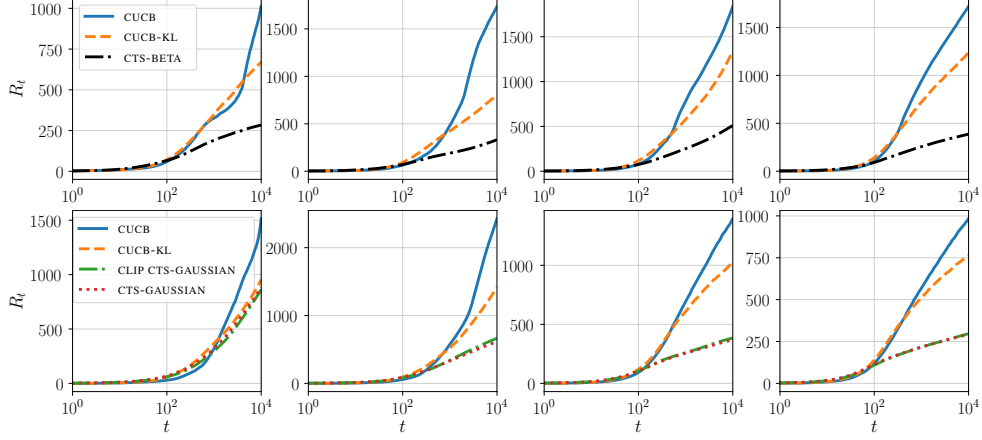

Figure 1: Cumulative regret (averaged over 50 simulations) for the shortest path problem. **Top:** with mutually independent outcomes, taking the opposite sum of means being $s = 70, 90, 110, 130$ respectively. **Bottom:** with correlated outcomes, taking the opposite sum of outcomes being $s = 70, 90, 110, 130$ respectively.

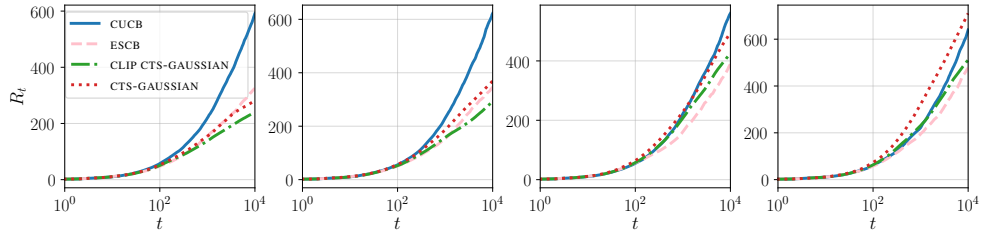

Figure 2: Cumulative regret (averaged over 50 simulations) for the matching problem with Gaussian outcomes, taking $c = -1/n, 0.2, 0.5, 1$ respectively.

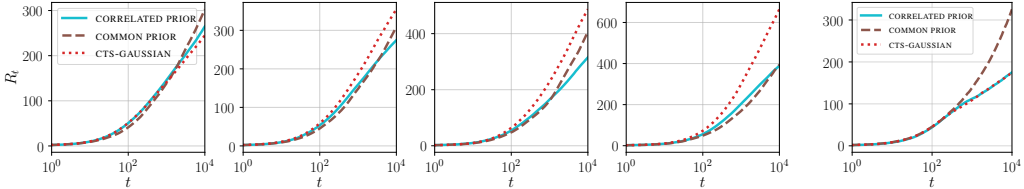

Figure 3: Comparison with correlated prior sampling and common prior sampling (averaged over 50 simulations). **The first 4:** for the $K_{4,4}$ matching problem, with Gaussian outcomes, taking $c = 0, 0.2, 0.5, 1$. **The last:** for $\mathcal{A} = \left\{ \{km + 1, \ldots, (k+1)m\} \mid k \in \left\{0, \ldots, \frac{n}{m} - 1\right\} \right\}, c = 0$.

## 6 Conclusion and future work

In this paper, we have provided the first efficient policies having an optimal regret bound for a wide spectrum of problems instances for CMAB with semi-bandit feedback. Our approach also answers the question of finding an analysis for CTS under correlated arm distributions. There are several possible extensions that could be considered as future work. For example, it would be interesting to have an analysis of CTS with a *correlated* (Gaussian) prior. Indeed, apart from the empirical gain, this would open up the possibility of estimating the covariance matrix and using it in the prior distribution. Further relevant results would be an analysis of CTS-BETA without the mutual independence of outcomes, or also an improved concentration bound for a sum of independent betas, relying on the kl rather than using sub-Gaussianity. This latter result would thus show that CTS-BETA dominates CUCB-KL, which is empirically observed.

## Broader Impact

This work does not present any foreseeable societal consequence.

## Acknowledgments and Disclosure of Funding

The research presented was supported by European CHIST-ERA project DELTA, French Ministry of Higher Education and Research, Nord-Pas-de-Calais Regional Council, French National Research Agency project BOLD (ANR19-CE23-0026-04).

It was also supported in part by a public grant as part of the Investissement d'avenir project, reference ANR-11-LABX-0056-LMH, LabEx LMH, in a joint call with Gaspard Monge Program for optimization, operations research and their interactions with data sciences.

## Footnotes

[1]We recall here the fact that in MAB, whether the horizon $T$ is known or not is not really relevant as algorithms can be easily adapted [Degenne and Perchet, 2016a].

[2]Henceforth, we typeset vectors in bold and indicate components with indices, i.e., $\mathbf{a} = (a_i)_{i \in [n]} \in \mathbb{R}^n$. We also let $\mathbf{e}_i$ be the $i^{th}$ canonical unit vector of $\mathbb{R}^n$, and define the incidence vector of any subset $A \subset [n]$ as $\mathbf{e}_A \triangleq \sum_{i \in A} \mathbf{e}_i$. We denote by $\mathbf{a} \odot \mathbf{b} \triangleq (a_i b_i)$ the Hadamard product of two vectors $\mathbf{a}$ and $\mathbf{b}$.

[3]In practice, for $\mathbf{C}$-sub Gaussian outcomes, the choice $\mathcal{N}\left( \overline{\boldsymbol{\mu}}_{t-1}, \left( C_{ij} N_{ij,t-1} N_{i,t-1}^{-1} N_{j,t-1}^{-1} \right)_{ij} \right)$ for the prior where $N_{ij,t-1} \triangleq \sum_{t' \in [t-1]} \mathbb{I}\{i \in A_{t'}\} \mathbb{I}\{j \in A_{t'}\}$ may be preferred.

[4]We also tried the policy (without displaying the results, for the sake of clarity) with covariance $\left(C_{ij} N_{i,t-1}^{-1/2} N_{j,t-1}^{-1/2}\right)_{ij}$, and observed about the same performance as the correlated prior approach.

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
