[Supplementary Material]

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

[5]This $D_i$ can be computed whenever linear maximization on $\mathcal{A}$ is efficient: for $x$ high enough, we have $\max_{A\in\mathcal{A}, \ i\in A}\sum_{j\in A}|C_{ij}| = C_{ii} - x + \max_{A\in\mathcal{A}}\sum_{j\in A}(|C_{ij}|\mathbb{I}\{j\neq i\} + x\mathbb{I}\{j=i\})$.

[6]We use the version that relies on Fatou's lemma (Durrett [2019], Theorem 5.7.6), so that it is not needed to have any additional condition on the stopping time $\tau^*$.

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

# A  Proof of Theorem 1

We first restate the complete non-asymptotic upper-bound as follows.

**Theorem.** *The policy $\pi$ described in Algorithm 1 has regret $R_T(\pi)$ bounded by*

$$16\log_2^2(16m)\sum_{i\in[n]}\frac{B^2\log(2^m|\mathcal{A}|T)}{\Delta_{i,\min}}+\Delta_{\max}(1+n)+\frac{nm^2\Delta_{\max}}{\left(\frac{\Delta_{\min}}{2B}-(m^{*2}+1)\varepsilon\right)^2}+\Delta_{\max}\frac{C}{\varepsilon^2}\left(\frac{C'}{\varepsilon^4}\right)^{m^*},$$

*where $C, C'$ are two universal constants, and $\varepsilon \in (0,1)$ is such that $\Delta_{\min}/(2B) - (m^{*2}+1)\varepsilon > 0$.*

## A.1  Preliminary lemmas

In order to prove Theorem 1, we modify two lemmas from Wang and Chen [2018]: first, in their Lemma 3, we replace $\varepsilon$ by $\Delta_{\min}/(2B) - (m^{*2}+1)\varepsilon > 0$, which gives the following Lemma 1.

**Lemma 1.** *In Algorithm 1, for any arm $i$, we have*

$$\mathbb{E}\left[\left|t\in[T],\ i\in A_t,\ |A_t|\cdot\left|\overline{\mu}_{i,t-1}-\mu_i^*\right| > \frac{\Delta_{\min}}{2B}-(m^{*2}+1)\varepsilon\right|\right]\le 1+\left(\frac{\Delta_{\min}}{2mB}-\frac{(m^{*2}+1)\varepsilon}{m}\right)^{-2}.$$

Then, we modify Lemma 4 from Wang and Chen [2018] as follows, leveraging on the mutual independence of $\theta_{1,t},\ldots,\theta_{n,t}$ to get a tighter confidence region for the sample $\boldsymbol{\theta}_t$.

**Lemma 2.** *In Algorithm 1, for all round $t$, we have*

$$\mathbb{P}\left[\left\|\mathbf{e}_{A_t}\odot\left(\boldsymbol{\theta}_t-\overline{\boldsymbol{\mu}}_{t-1}\right)\right\|_1\ge\sqrt{\frac{1}{2}\log(|\mathcal{A}|2^mT)\sum_{i\in A_t}\frac{1}{N_{i,t-1}}}\,\bigg|\,\mathcal{H}_t\right]\le 1/T.$$

*Proof.* From [Marchal et al., 2017], the Beta random variable from $\theta_{i,t}$ is sub-Gaussian with variance $1/(4N_{i,t-1})$. Thus, defining the functions

$$\alpha_t(A)\triangleq\sqrt{\frac{1}{2}\log(|\mathcal{A}|2^mT)\sum_{i\in A}\frac{1}{N_{i,t-1}}},\quad\text{and}\quad\lambda_t(A)\triangleq\frac{4\alpha_t(A)}{\sum_{i\in A}1/N_{i,t-1}},$$

we have

$$\begin{aligned}
\mathbb{P}\left[\left\|\mathbf{e}_{A_t}\odot\left(\boldsymbol{\theta}_t-\overline{\boldsymbol{\mu}}_{t-1}\right)\right\|_1\ge\alpha_t(A_t)\big|\mathcal{H}_t\right]&\le\sum_{A\in\mathcal{A}}\mathbb{P}\left[\left\|\mathbf{e}_A\odot\left(\boldsymbol{\theta}_t-\overline{\boldsymbol{\mu}}_{t-1}\right)\right\|_1\ge\alpha_t(A)\big|\mathcal{H}_t\right]\\
&\le\sum_{A\in\mathcal{A}}e^{-\lambda_t(A)\alpha_t(A)}\mathbb{E}\left[e^{\lambda_t(A)\left\|\mathbf{e}_A\odot\left(\boldsymbol{\theta}_t-\overline{\boldsymbol{\mu}}_{t-1}\right)\right\|_1}\Big|\mathcal{H}_t\right]\\
&\le\sum_{A\in\mathcal{A}}e^{-\lambda_t(A)\alpha_t(A)}\prod_{i\in A}\mathbb{E}\left[e^{\lambda_t(A)\left|\theta_{i,t}-\overline{\mu}_{i,t-1}\right|}\Big|\mathcal{H}_t\right]\\
&\le\sum_{A\in\mathcal{A}}e^{-\lambda_t(A)\alpha_t(A)}\prod_{i\in A}\mathbb{E}\left[e^{\lambda_t(A)\left(\theta_{i,t}-\overline{\mu}_{i,t-1}\right)}+e^{\lambda_t(A)\left(\overline{\mu}_{i,t-1}-\theta_{i,t}\right)}\Big|\mathcal{H}_t\right]\\
&\le\sum_{A\in\mathcal{A}}2^{|A|}e^{-\lambda_t(A)\alpha_t(A)}e^{\lambda_t(A)^2\sum_{i\in A}1/(8N_{i,t-1})}\le 1/T.
\end{aligned}$$

$\square$

## A.2  Main proof

With the two lemmas from the previous subsection, we are ready to demonstrate Theorem 1. We consider the following events.

- $\mathfrak{Z}_t\triangleq\{\Delta_t>0\}$
- $\mathfrak{B}_t\triangleq\left\{\exists i\in A_t,\ |A_t|\cdot\left|\overline{\mu}_{i,t-1}-\mu_i^*\right| > \Delta_{\min}/(2B)-(m^{*2}+1)\varepsilon\right\}$

- $\mathfrak{C}_t \triangleq \left\{ \left\| \mathbf{e}_{A_t} \odot (\boldsymbol{\theta}_t - \boldsymbol{\mu}^*) \right\|_1 > \Delta_t/B - \left(m^{*2} + 1\right)\varepsilon \right\}$
- $\mathfrak{D}_t \triangleq \left\{ \left\| \mathbf{e}_{A_t} \odot (\boldsymbol{\theta}_t - \overline{\boldsymbol{\mu}}_{t-1}) \right\|_1 \geq \sqrt{0.5 \cdot \log(|\mathcal{A}|2^m T) \sum_{i \in A_t} 1/N_{i,t-1}} \right\}$.

We break down our analysis into 4 steps. The main novelties are in the last two steps: Step 3 gives us the tighter dependence in $m$, and Step 4, that contains the main difficulties, gives the new exponential constant term.

**Step 1: bound under $\mathfrak{Z}_t \wedge \mathfrak{B}_t$**  By Lemma 1,

$$\sum_{t \in [T]} \mathbb{E}[\Delta_t \mathbb{I}\{\mathfrak{Z}_t \wedge \mathfrak{B}_t\}] \leq \Delta_{\max} \sum_{i \in [n]} \mathbb{E}\left[ \left| t \in [T], \ i \in A_t, \ |A_t| \cdot \left| \overline{\mu}_{i,t-1} - \mu_i^* \right| > \Delta_{\min}/(2B) - (m^{*2}+1)\varepsilon \right| \right]$$

$$\leq n\Delta_{\max}\left( 1 + \left( \frac{\Delta_{\min}}{2mB} - \frac{(m^{*2}+1)\varepsilon}{m} \right)^{-2} \right).$$

**Step 2: bound under $\mathfrak{Z}_t \wedge \neg\mathfrak{B}_t \wedge \mathfrak{C}_t \wedge \mathfrak{D}_t$**  By Lemma 2,

$$\sum_{t \in [T]} \mathbb{E}[\Delta(A_t)\mathbb{I}\{\mathfrak{Z}_t \wedge \neg\mathfrak{B}_t \wedge \mathfrak{C}_t \wedge \mathfrak{D}_t\}] \leq \Delta_{\max} \sum_{t \in [T]} \mathbb{E}[\mathbb{P}[\mathfrak{D}_t|\mathcal{H}_t]] \leq \Delta_{\max} \sum_{t \in [T]} 1/T = \Delta_{\max}.$$

**Step 3: bound under $\mathfrak{Z}_t \wedge \neg\mathfrak{B}_t \wedge \mathfrak{C}_t \wedge \neg\mathfrak{D}_t$**

$$\begin{aligned}
\Delta_t/B &\leq \left\| \mathbf{e}_{A_t} \odot (\boldsymbol{\theta}_t - \boldsymbol{\mu}^*) \right\|_1 + \left( m^{*2} + 1 \right)\varepsilon & \mathfrak{C}_t \\
&\leq \left\| \mathbf{e}_{A_t} \odot (\boldsymbol{\theta}_t - \overline{\boldsymbol{\mu}}_{t-1}) \right\|_1 + \left\| \mathbf{e}_{A_t} \odot (\overline{\boldsymbol{\mu}}_{t-1} - \boldsymbol{\mu}^*) \right\|_1 + \left( m^{*2} + 1 \right)\varepsilon \\
&\leq \left\| \mathbf{e}_{A_t} \odot (\boldsymbol{\theta}_t - \overline{\boldsymbol{\mu}}_{t-1}) \right\|_1 + \Delta_{\min}/(2B) - \left( m^{*2} + 1 \right)\varepsilon + \left( m^{*2} + 1 \right)\varepsilon & \neg\mathfrak{B}_t \\
&\leq \left\| \mathbf{e}_{A_t} \odot (\boldsymbol{\theta}_t - \overline{\boldsymbol{\mu}}_{t-1}) \right\|_1 + \Delta_t/(2B) & \mathfrak{Z}_t \\
&\leq \sqrt{\frac{1}{2}\log(|\mathcal{A}|2^m T) \sum_{i \in A_t} \frac{1}{N_{i,t-1}}} + \Delta_t/(2B). & \neg\mathfrak{D}_t
\end{aligned}$$

So we have that the following event holds

$$\mathfrak{A}_t \triangleq \left\{ \Delta_t \leq B\sqrt{2\log(|\mathcal{A}|2^m T) \sum_{i \in A_t} \frac{1}{N_{i,t-1}}} \right\}.$$

We can thus apply Theorem 4 (see Appendix E) to get the bound

$$\begin{aligned}
\sum_{t \in [T]} \mathbb{E}[\Delta_t \mathbb{I}\{\mathfrak{Z}_t, \neg\mathfrak{B}_t, \mathfrak{C}_t, \neg\mathfrak{D}_t\}] &\leq \sum_{t \in [T]} \mathbb{E}[\Delta_t \mathbb{I}\{\mathfrak{A}_t\}] \\
&\leq 32B^2 \log_2^2(4\sqrt{m}) \sum_{i \in [n]} \Delta_{i,\min}^{-1} 2\log(|\mathcal{A}|2^m T).
\end{aligned}$$

**Step 4: bound under $\mathfrak{Z}_t \wedge \neg\mathfrak{C}_t$**  We consider the following events for a subset $Z \subset [n]$

$$\mathfrak{R}(\boldsymbol{\theta}', Z) \triangleq \left\{ Z \subset \mathrm{Oracle}(\boldsymbol{\theta}'), \ \left\| \mathbf{e}_{\mathrm{Oracle}(\boldsymbol{\theta}')} \odot (\boldsymbol{\theta}' - \boldsymbol{\mu}^*) \right\|_1 > \Delta\big(\mathrm{Oracle}(\boldsymbol{\theta}')\big) - (k^{*2}+1)\varepsilon \right\}$$

$$\mathfrak{S}_t(Z) \triangleq \left\{ \forall \boldsymbol{\theta}' \text{ s.t. } \left\| (\boldsymbol{\mu}^* - \boldsymbol{\theta}') \odot \mathbf{e}_Z \right\|_\infty \leq \varepsilon, \ \mathfrak{R}(\boldsymbol{\theta}' \odot \mathbf{e}_Z + \boldsymbol{\theta}_t \odot \mathbf{e}_{Z^c}, Z) \text{ holds} \right\} \qquad (3)$$

$$\mathfrak{T}_t(Z) \triangleq \left\{ \left\| (\boldsymbol{\mu}^* - \boldsymbol{\theta}_t) \odot \mathbf{e}_Z \right\|_\infty > \varepsilon \right\}.$$

We can state the three following lemmas. Note that Lemma 3 is exactly the Lemma 1 from Wang and Chen [2018]. The other two replace their Lemma 7.

**Lemma 3.** *In Algorithm 1, for all round t, we have*

$$\mathfrak{Z}_t, \neg\mathfrak{C}_t \Rightarrow \exists Z \subset A^*, \ Z \neq \emptyset \ \text{s.t. the event } \mathfrak{S}_t(Z) \wedge \mathfrak{T}_t(Z) \text{ holds.}$$

**Lemma 4.** *Given $Z \subset A^*$, $Z \neq \emptyset$, let $\tau_q$ be the round at which $\mathfrak{S}_t(Z) \wedge \neg\mathfrak{T}_t(Z)$ occurs for the $q$-th time, and let $\tau_0 = 0$. Then, in Algorithm 1, we have*

$$\mathbb{E}\left[\sum_{t=\tau_q+1}^{\tau_{q+1}} \mathbb{I}\{\mathfrak{S}_t(Z), \mathfrak{T}_t(Z)\}\right] \leq \mathbb{E}\left[\sup_{\tau \geq \tau_q+1} \prod_{i \in Z} \frac{1}{\mathbb{P}[|\theta_{i,\tau} - \mu_i^*| \leq \varepsilon | \mathcal{H}_\tau]}\right] - 1.$$

**Lemma 5.** *In Algorithm 1, we have*

$$\mathbb{E}\left[\sup_{\tau \geq \tau_q+1} \prod_{i \in Z} \frac{1}{\mathbb{P}[|\theta_{i,\tau} - \mu_i^*| \leq \varepsilon | \mathcal{H}_\tau]}\right] - 1 \leq \begin{cases} \left(c\varepsilon^{-4}\right)^{|Z|} & \text{for every } q \geq 0 \\ e^{-\varepsilon^2 q/8}\left(c'\varepsilon^{-4}\right)^{|Z|} & \text{if } q > 8/\varepsilon^2, \end{cases}$$

*where $c$ and $c'$ are two universal constants.*

These lemmas allow us to get a constant regret under the event $\mathfrak{Z}_t \wedge \neg\mathfrak{C}_t$. Indeed, we have from Lemma 3 that

$$\sum_{t \in [T]} \mathbb{E}[\Delta_t \mathbb{I}\{\mathfrak{Z}_t \wedge \neg\mathfrak{C}_t\}] \leq \Delta_{\max} \sum_{Z \subset A^*, \ Z \neq \emptyset} \mathbb{E}\left[\sum_{t \in [T]} \mathbb{I}\{\mathfrak{S}_t(Z) \wedge \mathfrak{T}_t(Z)\}\right]$$

$$= \Delta_{\max} \sum_{Z \subset A^*, \ Z \neq \emptyset} \sum_{q \geq 0} \mathbb{E}\left[\sum_{t=\tau_q+1}^{\tau_{q+1}} \mathbb{I}\{\mathfrak{S}_t(Z), \mathfrak{T}_t(Z)\}\right].$$

Lemma 4 and 5 gives that the above is further upper bounded by

$$\Delta_{\max} \sum_{Z \subset A^*, \ Z \neq \emptyset} \left(\sum_{q=0}^{\lceil 8/\varepsilon^2 \rceil - 1} \left(c\varepsilon^{-4}\right)^{|Z|} + \sum_{q \geq \lceil 8/\varepsilon^2 \rceil} e^{-\varepsilon^2 q/8}\left(c'\varepsilon^{-4}\right)^{|Z|}\right)$$

which is bounded by

$$\Delta_{\max} \frac{C}{\varepsilon^2}\left(\frac{C'}{\varepsilon^4}\right)^{m^*},$$

where $C$ and $C'$ are two universal constants. This concludes the proof of the theorem.

*Proof of Lemma 4.* Since $\mathfrak{S}_t(Z), \mathfrak{T}_t(Z)$ are independent conditioned on the history $\mathcal{H}_t$, the LHS is

$$\mathbb{E}\left[\sum_{k \geq 1}(k-1)\mathbb{P}\left[\neg\mathfrak{T}_{t_{k,q}}(Z)\big|\mathcal{H}_{t_{k,q}}\right]\prod_{j=1}^{k-1}\mathbb{P}\left[\mathfrak{T}_{t_{j,q}}(Z)\big|\mathcal{H}_{t_{j,q}}\right]\right],$$

where $t_{k,q}$ is the round $t$ where $\mathfrak{S}_t(Z)$ holds for the $k$-th time since the beginning of the round $\tau_q + 1$. Within the expectation, one can recognize the expectation of a time-varying geometric distribution, where the success probability of the $k$-th trial is $\mathbb{P}\left[\neg\mathfrak{T}_{t_{k,q}}(Z)\big|\mathcal{H}_{t_{k,q}}\right]$. We can upper bound this inner expectation by the expectation of a geometric distribution whose success probability

$$\inf_{\tau \geq \tau_q+1} \mathbb{P}[\neg\mathfrak{T}_\tau(Z)|\mathcal{H}_\tau] = \inf_{\tau \geq \tau_q+1} \prod_{i \in Z} \mathbb{P}[|\theta_{i,\tau} - \mu_i^*| \leq \varepsilon | \mathcal{H}_\tau]$$

is lower than all the success probabilities of the time-varying geometric distribution. This gives the result by monotonicity of the expectation, and rewriting the expectation of the geometric distribution. $\square$

*Proof of Lemma 5.* For any arm $i \in [n]$, $k_i \in \mathbb{N}$, we define $p_{i,k_i}$ as the probability of $\left|\widetilde{\theta}_{i,k_i} - \mu_i^*\right| \leq \varepsilon$, where $\widetilde{\theta}_{i,k_i}$ is a sample from the posterior of arm $i$ when there are $k_i$ observations of arm $i$ (i.e., $p_{i,k_i}$ is a random variable measurable with respect to those $k_i$ independent draws of arm $i$). From Lemma 5,6 in Wang and Chen [2018], we know that

$$\mathbb{E}\left[\frac{1}{p_{i,k_i}}\right] \leq \begin{cases} 4/\varepsilon^2 & \text{for every } k_i \geq 0 \\ 1 + 6c'' \cdot e^{-\varepsilon^2 k_i/2}\varepsilon^{-2} + \frac{2}{e^{\varepsilon^2 k_i/8} - 2} & \text{if } k_i > 8/\varepsilon^2, \end{cases}$$

for some universal constant $c''$. Since $\mathfrak{S}_t(Z) \wedge \neg\mathfrak{T}_t(Z)$ implies that $Z \subset A_t$, we know that for $\tau \geq \tau_q + 1$, $N_{i,\tau-1} \geq q$ for all $i \in Z$. Using the mutual independence of outcomes, and the fact that the distribution of $\theta_{i,\tau}$ depends only on the history of arm $i$, we have

$$\mathbb{E}\left[\sup_{\tau \geq \tau_q+1} \prod_{i \in Z} \frac{1}{\mathbb{P}[\,|\theta_{i,\tau} - \mu_i^*| \leq \varepsilon|\mathcal{H}_\tau]}\right] - 1$$

$$= \mathbb{E}\left[\sup_{\tau \geq \tau_q+1} \sum_{Z' \subset Z, \, Z' \neq \emptyset} \prod_{i \in Z'} \left(\frac{1}{\mathbb{P}[\,|\theta_{i,\tau} - \mu_i^*| \leq \varepsilon|\mathcal{H}_\tau]} - 1\right)\right]$$

$$\leq \sum_{Z' \subset Z, \, Z' \neq \emptyset} \mathbb{E}\left[\prod_{i \in Z'} \sup_{\tau \geq \tau_q+1} \left(\frac{1}{\mathbb{P}[\,|\theta_{i,\tau} - \mu_i^*| \leq \varepsilon|\mathcal{H}_\tau]} - 1\right)\right]$$

$$\leq \sum_{Z' \subset Z, \, Z' \neq \emptyset} \mathbb{E}\left[\prod_{i \in Z'} \sum_{k_i \geq q} \left(\frac{1}{p_{i,k_i}} - 1\right)\right],$$

$$= \sum_{Z' \subset Z, \, Z' \neq \emptyset} \prod_{i \in Z'} \mathbb{E}\left[\sum_{k_i \geq q} \left(\frac{1}{p_{i,k_i}} - 1\right)\right].$$

From this point, there are two cases: If $q > 8/\varepsilon^2$,

$$\leq \sum_{Z' \subset Z, \, Z' \neq \emptyset} \prod_{i \in Z'} \sum_{k_i \geq q} \left(6c'' \cdot e^{-\varepsilon^2 k/2}\varepsilon^{-2} + 2e^{-\varepsilon^2 k/8}\left(1 - 2e^{-\varepsilon^2 k/8}\right)^{-1}\right)$$

$$\leq \sum_{Z' \subset Z, \, Z' \neq \emptyset} \prod_{i \in Z'} \left(6c'' \cdot e^{-\varepsilon^2 q/2}\varepsilon^{-2} \sum_{k \geq 0} e^{-\varepsilon^2 k/2} + 2e^{-\varepsilon^2 q/8}\left(1 - 2e^{-\varepsilon^2 q/8}\right)^{-1} \sum_{k \geq 0} e^{-\varepsilon^2 k/8}\right)$$

$$= \sum_{Z' \subset Z, \, Z' \neq \emptyset} \prod_{i \in Z'} \left(6c'' \cdot e^{-\varepsilon^2 q/2}\varepsilon^{-2}\left(1 - e^{-\varepsilon^2/2}\right)^{-1} + 2e^{-\varepsilon^2 q/8}\left(1 - 2e^{-\varepsilon^2 q/8}\right)^{-1}\left(1 - e^{-\varepsilon^2/8}\right)^{-1}\right)$$

$$\leq \sum_{Z' \subset Z, \, Z' \neq \emptyset} \prod_{i \in Z'} \left(6c'' \cdot e^{-\varepsilon^2 q/2}\varepsilon^{-2} \cdot 2\varepsilon^{-2}\left(1 - e^{-1/2}\right)^{-1} + 2e^{-\varepsilon^2 q/8}\left(1 - 2e^{-1}\right)^{-1} \cdot 8\varepsilon^{-2}\left(1 - e^{-1/8}\right)^{-1}\right)$$

$$\leq \sum_{Z' \subset Z, \, Z' \neq \emptyset} e^{-|Z'|\varepsilon^2 q/8}\left(12c'' \cdot e^{-3}\left(1 - e^{-1/2}\right)^{-1} \cdot \varepsilon^{-4} + 16\left(1 - 2e^{-1}\right)^{-1}\varepsilon^{-2}\left(1 - e^{-1/8}\right)^{-1}\right)^{|Z'|}$$

$$\leq e^{-\varepsilon^2 q/8}\left(12c'' \cdot e^{-3}\left(1 - e^{-1/2}\right)^{-1}\varepsilon^{-4} + 16\left(1 - 2e^{-1}\right)^{-1}\varepsilon^{-2}\left(1 - e^{-1/8}\right)^{-1} + 1\right)^{|Z|}$$

$$\leq e^{-\varepsilon^2 q/8}\left(c'\varepsilon^{-4}\right)^{|Z|},$$

and if $q \leq 8/\varepsilon^2$,

$$\leq \sum_{Z' \subset Z, \, Z' \neq \emptyset} \prod_{i \in Z'} \left(\sum_{k=q}^{\lfloor 8/\varepsilon^2 \rfloor} (4/\varepsilon^2 - 1) + \sum_{k \geq \lfloor 8/\varepsilon^2 \rfloor + 1}^{\infty} \left(6c \cdot e^{-\varepsilon^2 k/2}\varepsilon^{-2} + 2e^{-\varepsilon^2 k/8}\left(1 - 2e^{-\varepsilon^2 k/8}\right)^{-1}\right)\right)$$

$$\leq \sum_{Z' \subset Z, \, Z' \neq \emptyset} \prod_{i \in Z'} \left(36\varepsilon^{-4} + 12c \cdot e^{-4}\left(1 - e^{-1/2}\right)^{-1}\varepsilon^{-4} + 16e^{-1}\left(1 - 2e^{-1}\right)^{-1}\varepsilon^{-2}\left(1 - e^{-1/8}\right)^{-1}\right)$$

$$\leq \left(c\varepsilon^{-4}\right)^{|Z|},$$

where $c, c'$ are two universal constant. $\qquad\qquad\qquad\qquad\qquad\qquad\qquad\qquad$ □

### A.3 Discussion on the new exponential constant term (step 4 in the above proof)

We give here an explanation concerning the modification of Lemma 7 from Wang and Chen [2018]. First, we respectfully disagree with the end of their proof, where the expected number of time slots for $\mathfrak{S}_t(Z) \wedge \neg\mathfrak{T}_t(Z)$ to occur is a weighted mean of expectations where the counters are fixed and non-random. To obtain such a weighted mean, they have conditioned on the value of the counters. However, counters depend on the chosen action, and thus on the outcomes previously obtained, so conditioning on it would modify the expectation, since the term inside the expectation not only depends on counters, but also on outcomes obtained so far. To illustrate more clearly this point, let us focus on one arm $i$, and consider the extreme case where we get a new sample (i.e. the counter is incremented) only if samples $Y_{i,t}$ previously obtained from $i$ were all 0, say. Then conditioning on the fact that the counter is incremented would remove all the randomness of samples $Y_{i,t}$, and we thus can't consider an expectation on those samples as if their randomness was not impacted.

We now expose our approach to overcome this issue. We first rewrite the above mentioned expected number of time slots as the expectation (over the history) of the expectation of a time-varying geometric distribution, where the time-varying success probability depends on the history. The inner expectation can be bounded by the expectation of a geometric distribution whose success probability is the infimum over all the success probabilities of the time-varying geometric distribution. Let's note that this gives us the inverse success probability minus one, as in Wang and Chen [2018], but that counters are still random. We use that this inverse probability can be factorized: from the relation $\prod_{i\in A} a_i - 1 = \sum_{A'\subset A, \ A'\neq\emptyset} \prod_{i\in A'}(a_i - 1)$, valid for any vector $\mathbf{a} = (a_i)$ on a set $A$, and from the mutual independence of outcomes, we're reduced to bounding the expectation in the one-dimensional case. To overcome the randomness of the counters, we use an union bound. It is this union bound that brings a larger dependence on the constant term, because it forces us to look at a sum of the form $\sum_q \sum_{k\geq q} x_k$, instead of a simply $\sum_q x_q$. Let's remark that Wang and Chen [2018] use the eventual exponential decreasing of the sequence $(x_q)$ in order to get their final bound. We manage to deal with the sequence $\left(\sum_{k\geq q} x_k\right)$ instead, by noticing that the eventual exponential decreasing of the sequence $(x_q)$ implies the eventual exponential decreasing of the sequence $\left(\sum_{k\geq q} x_k\right)$.

## B  Proof of Proposition 1

Assumption 4 encompasses $\kappa_i^2$-sub Gaussian outcomes with $D_i = \kappa_i^2 m$ for all $i \in [n]$. Indeed, let $\boldsymbol{\lambda} = \boldsymbol{\lambda} \odot \mathbf{e}_A$ for some action $A$ and observe that

$$\mathbb{E}\left[e^{\boldsymbol{\lambda}^{\top}(\mathbf{X}-\boldsymbol{\mu}^*)}\right] \leq \mathbb{E}\left[\sum_i \frac{|\kappa_i\lambda_i|}{\|\boldsymbol{\kappa}\odot\boldsymbol{\lambda}\|_1} e^{\|\boldsymbol{\kappa}\odot\boldsymbol{\lambda}\|_1 \mathrm{sign}(\lambda_i)\frac{X_i-\mu_i^*}{\kappa_i}}\right] \leq e^{\|\boldsymbol{\kappa}\odot\boldsymbol{\lambda}\|_1^2/2} \leq e^{\|\boldsymbol{\kappa}\odot\boldsymbol{\lambda}\|_2^2|A|/2} \leq e^{\|\boldsymbol{\kappa}\odot\boldsymbol{\lambda}\|_2^2 m/2}.$$

The case of $\mathbf{C}$-sub-Gaussian outcomes with a known sub-Gaussian matrix $\mathbf{C}$ (i.e., $\mathbb{E}\left[e^{\boldsymbol{\lambda}^{\top}(\mathbf{X}-\boldsymbol{\mu}^*)}\right] \leq e^{\boldsymbol{\lambda}^{\top}\mathbf{C}\boldsymbol{\lambda}/2}$ for all $\boldsymbol{\lambda} \in \mathbb{R}^n$) is also captured, taking[5] $D_i = \max_{A\in\mathcal{A}, \ i\in A}\sum_{j\in A}|C_{ij}|$. Indeed, for an action $A$,

$$\sum_{i,j\in A}\lambda_i\lambda_j C_{ij} \leq \sum_{i,j\in A}\frac{\lambda_i^2 + \lambda_j^2}{2}|C_{ij}| = \sum_{i\in A}\lambda_i^2\sum_{j\in A}|C_{ij}| \leq \sum_{i\in n}\lambda_i^2 \max_{A\in\mathcal{A}, \ i\in A}\sum_{j\in A}|C_{ij}|.$$

## C  Proof of Theorem 2

We beginning by stating the complete version of Theorem 2.

**Theorem.** *The policy $\pi$ described in Algorithm 2 has regret $R_T(\pi)$ bounded by*

$$256\log_2^2(4\sqrt{m})\sum_{i\in[n]}\frac{B^2\beta D_i\log(2^m|\mathcal{A}|T)}{\Delta_{i,\min}}+\Delta_{\max}(1+2n)$$

$$+\frac{nm^2\Delta_{\max}}{\left(\frac{\Delta_{\min}}{2B}-(m^{*2}+1)\varepsilon\right)^2}+\Delta_{\max}\left(C\varepsilon^{-2}\beta\max_i D_i\right)\left(\frac{C'}{\sqrt{\beta-1}}\varepsilon^{-4}\beta^3\max_i D_i^2\right)^{m^*},$$

*where $C,C'$ are two universal constants, and $\varepsilon\in(0,1)$ is such that $\Delta_{\min}/(2B)-(m^{*2}+1)\varepsilon>0$.*

For the proof of Theorem 2, we consider the same events as in the proof of Theorem 1, except for the event $\mathfrak{D}_t$, that becomes

$$\mathfrak{D}_t\triangleq\left\{\left\|\mathbf{e}_{A_t}\odot\left(\boldsymbol{\theta}_t-\overline{\boldsymbol{\mu}}_{t-1}\right)\right\|_1\geq\sqrt{2\log(|\mathcal{A}|2^mT)\sum_{i\in A_t}\beta D_i/N_{i,t-1}}\right\}.$$

Step 1 is unchanged. Step 2 and Step 3 are modified only through the event $\mathfrak{D}_t$, using the following modification of Lemma 2.

**Lemma 6.** *In Algorithm 2, for all round t, we have that $\mathbb{P}[\mathfrak{D}_t|\mathcal{H}_t]\leq 1/T$.*

*Proof.* We rely on the fact that conditionally on the history, the sample $\boldsymbol{\theta}_t$ is Gaussian of mean $\overline{\boldsymbol{\mu}}_{t-1}$ and of diagonal covariance given by $\beta D_i N_{i,t-1}^{-1}$. We thus define the functions

$$\alpha_t(A)\triangleq\sqrt{2\log(|\mathcal{A}|2^mT)\sum_{i\in A}\frac{\beta D_i}{N_{i,t-1}}},\quad\text{and}\quad\lambda_t(A)\triangleq\frac{\alpha_t(A)}{\sum_{i\in A}\beta D_i/N_{i,t-1}},$$

we have

$$\mathbb{P}\left[\left\|\mathbf{e}_{A_t}\odot\left(\boldsymbol{\theta}_t-\overline{\boldsymbol{\mu}}_{t-1}\right)\right\|_1\geq\alpha_t(A_t)\big|\mathcal{H}_t\right]\leq\sum_{A\in\mathcal{A}}\mathbb{P}\left[\left\|\mathbf{e}_A\odot\left(\boldsymbol{\theta}_t-\overline{\boldsymbol{\mu}}_{t-1}\right)\right\|_1\geq\alpha_t(A)\big|\mathcal{H}_t\right]$$

$$\leq\sum_{A\in\mathcal{A}}e^{-\lambda_t(A)\alpha_t(A)}\mathbb{E}\left[e^{\lambda_t(A)\left\|\mathbf{e}_A\odot\left(\boldsymbol{\theta}_t-\overline{\boldsymbol{\mu}}_{t-1}\right)\right\|_1}\Big|\mathcal{H}_t\right]$$

$$\leq\sum_{A\in\mathcal{A}}e^{-\lambda_t(A)\alpha_t(A)}\prod_{i\in A}\mathbb{E}\left[e^{\lambda_t(A)\left|\theta_{i,t}-\overline{\mu}_{i,t-1}\right|}\Big|\mathcal{H}_t\right]$$

$$\leq\sum_{A\in\mathcal{A}}e^{-\lambda_t(A)\alpha_t(A)}\prod_{i\in A}\mathbb{E}\left[e^{\lambda_t(A)\left(\theta_{i,t}-\overline{\mu}_{i,t-1}\right)}+e^{\lambda_t(A)\left(\overline{\mu}_{i,t-1}-\theta_{i,t}\right)}\Big|\mathcal{H}_t\right]$$

$$\leq\sum_{A\in\mathcal{A}}2^{|A|}e^{-\lambda_t(A)\alpha_t(A)}e^{\lambda_t(A)^2\sum_{i\in A}\beta D_i/(2N_{i,t-1})}\leq 1/T.$$

$\square$

The final bound on the regret in Step 3 is obtained using the same derivation as in Theorem 1, which gives the following leading term:

$$256\log_2^2(4\sqrt{m})\sum_{i\in[n]}\frac{B^2\beta D_i\log(2^m|\mathcal{A}|T)}{\Delta_{i,\min}}.$$

In the following, we consider the last step, consisting in bounding the regret under the event $\mathfrak{Z}_t$ and $\neg\mathfrak{C}_t$. From the initialization phase, we also assume that the event

$$\mathfrak{M}_t\triangleq\{\forall i\in[n],\ N_{i,t-1}\geq 1\}$$

holds (the regret under the complementary event is clearly bounded by $n\Delta_{\max}$). If there is no initialization, we can have $q=0$ in the following, noticing that when $\theta_{i,t}$ is uniform on $[a,b]$, then the probability $\mathbb{P}[|\theta_{i,t}-\mu_i^*|\leq\varepsilon|\mathcal{H}_t]$ is equal to $2\varepsilon/(b-a)$.

**Step 4: bound under $\mathfrak{M}_t \wedge \mathfrak{Z}_t \wedge \neg \mathfrak{C}_t$**   We use the independence of the prior, as for Theorem 1, to obtain the following upper bound, using $\mathfrak{M}_t$ to be able to start from $q = 1$.

$$
\sum_{t \in [T]} \mathbb{E}[\Delta(A_t) \mathbb{I}\{\mathfrak{M}_t \wedge \mathfrak{Z}_t \wedge \neg \mathfrak{C}_t\}] \leq \sum_{Z \subset A^*,\, Z \neq \emptyset} \sum_{q \geq 1} \mathbb{E}\left[ \sup_{\tau \geq \tau_q + 1} \sum_{Z' \subset Z,\, Z' \neq \emptyset} \prod_{i \in Z'} \left( \frac{1}{\mathbb{P}[|\theta_{i,\tau} - \mu_i^*| \leq \varepsilon | \mathcal{H}_\tau]} - 1 \right) \right]
$$

$$
\leq \underbrace{\sum_{Z \subset A^*,\, Z \neq \emptyset} \sum_{q \geq 1} \sum_{Z' \subset Z,\, Z' \neq \emptyset} \mathbb{E}\left[ \sup_{\tau \geq \tau_q + 1} \prod_{i \in Z'} \left( \frac{1}{\mathbb{P}[|\theta_{i,\tau} - \mu_i^*| \leq \varepsilon | \mathcal{H}_\tau]} - 1 \right) \right]}_{(4)}.
$$

However, the expectation can't be put inside the product since outcomes are not mutually independent. We can still take a union bound on counters:

$$
(4) \leq \sum_{Z' \subset Z,\, Z' \neq \emptyset} \sum_{\mathbf{k} \in [q..\infty)^{Z'}} \mathbb{E}\left[ \sup_{\tau \geq \tau_q + 1} \mathbb{I}\{\forall i \in Z',\ N_{i,\tau-1} = k_i\} \prod_{i \in Z'} \left( \frac{1}{\mathbb{P}[|\theta_{i,\tau} - \mu_i^*| \leq \varepsilon | \mathcal{H}_\tau]} - 1 \right) \right].
$$

One can notice that for all $i \in Z'$, all $k_i \geq q$, $\mathbb{I}\{N_{i,\tau-1} = k_i\}\left( \frac{1}{\mathbb{P}[|\theta_{i,\tau} - \mu_i^*| \leq \varepsilon | \mathcal{H}_\tau]} - 1 \right)$ is of the form $\mathbb{I}\{N_{i,\tau-1} = k_i\} g_i\left(\left|\overline{\mu}_{i,\tau-1} - \mu_i^*\right|\right)$, with $g_i$ being an increasing function on $\mathbb{R}_+$. Indeed, we see that the conditional distribution of $\theta_{i,\tau} - \overline{\mu}_{i,\tau-1}$ is $\mathcal{N}\left(0, \beta D_i N_{i,\tau-1}^{-1}\right)$, which is symmetric, so we have

$$
\mathbb{P}[|\theta_{i,\tau} - \mu_i^*| \leq \varepsilon | \mathcal{H}_\tau] = \mathbb{P}\left[\left|\theta_{i,\tau} - \overline{\mu}_{i,\tau-1} + \left|\overline{\mu}_{i,\tau-1} - \mu_i^*\right|\right| \leq \varepsilon \big| \mathcal{H}_\tau\right].
$$

In addition, under $\mathbb{I}\{N_{i,\tau-1} = k_i\}$, the conditional distribution of $\theta_{i,\tau} - \overline{\mu}_{i,\tau-1}$ does not depend on the history, but only on $k_i$. Therefore, the above probability is a function of $\left|\overline{\mu}_{i,\tau-1} - \mu_i^*\right|$ and so the function $g_i$ exists. It is increasing on $\mathbb{R}_+$ because for any fixed $\sigma > 0$,

$$
\frac{\partial}{\partial x} \int_{x-\varepsilon}^{x+\varepsilon} \frac{1}{\sqrt{2\pi\sigma^2}} e^{-\frac{u^2}{2\sigma^2}} \mathrm{d}u = \frac{1}{\sqrt{2\pi\sigma^2}} \left( e^{-\frac{(x+\varepsilon)^2}{2\sigma^2}} - e^{-\frac{(x-\varepsilon)^2}{2\sigma^2}} \right) < 0 \text{ for } x > 0.
$$

In particular, we can consider the inverse function $g_i^{-1}$. We now want to use a stochastic dominance argument in order to treat the outcomes as if they were Gaussian: we have for any $\mathbf{k} \in [q..\infty)^{Z'}$,

$$
\mathbb{E}\left[ \sup_{\tau \geq \tau_q + 1} \prod_{i \in Z'} \left( \mathbb{I}\{N_{i,\tau-1} = k_i\} g_i\left(\left|\overline{\mu}_{i,\tau-1} - \mu_i^*\right|\right) \right) \right]
$$

$$
= \mathbb{E}\left[ \sup_{\tau \geq \tau_q + 1} \prod_{i \in Z'} \left( \mathbb{I}\{N_{i,\tau-1} = k_i\} \int_0^\infty \mathbb{I}\{g_i\left(\left|\overline{\mu}_{i,\tau-1} - \mu_i^*\right|\right) \geq u_i\} \mathrm{d}u_i \right) \right]
$$

$$
\leq \int_{\mathbf{u} \in \mathbb{R}_+^{Z'}} \mathbb{E}\left[ \sup_{\tau \geq \tau_q + 1} \prod_{i \in Z'} \mathbb{I}\{N_{i,\tau-1} = k_i\} \mathbb{I}\{g_i\left(\left|\overline{\mu}_{i,\tau-1} - \mu_i^*\right|\right) \geq u_i\} \right] \mathrm{d}\mathbf{u}
$$

$$
= \int_{\mathbf{u} \in \mathbb{R}_+^{Z'}} \mathbb{E}\left[ \prod_{i \in Z'} \mathbb{I}\{N_{i,\tau^*-1} = k_i\} \mathbb{I}\{g_i\left(\left|\overline{\mu}_{i,\tau^*-1} - \mu_i^*\right|\right) \geq u_i\} \right] \mathrm{d}\mathbf{u}, \qquad (5)
$$

where $\tau^*$ is the first time $\tau$ such that the event $\mathbb{I}\{\forall i \in Z',\ N_{i,\tau-1} = k_i$ and $g_i\left(\left|\overline{\mu}_{i,\tau-1} - \mu_i^*\right|\right) \geq u_i\}$ holds, and is $\infty$ if it never holds.

$$
(5) = \int_{\mathbf{u} \in \mathbb{R}_+^{Z'}} \mathbb{E}\left[ \prod_{i \in Z'} \mathbb{I}\{N_{i,\tau^*-1} = k_i\} \mathbb{I}\{g_i\left(\left|\overline{\mu}_{i,\tau^*-1} - \mu_i^*\right|\right) \geq u_i \vee g_i(0)\} \right] \mathrm{d}\mathbf{u}
$$

$$
= \int_{\mathbf{u} \in \mathbb{R}_+^{Z'}} \mathbb{E}\left[ \prod_{i \in Z'} \mathbb{I}\{N_{i,\tau^*-1} = k_i\} \mathbb{I}\{\left|\overline{\mu}_{i,\tau^*-1} - \mu_i^*\right| \geq g_i^{-1}(u_i \vee g_i(0))\} \right] \mathrm{d}\mathbf{u}
$$

$$
= \int_{\mathbf{u} \in \mathbb{R}_+^{Z'}} \sum_{\mathbf{s} \in \{-1,1\}^{Z'}} \underbrace{\mathbb{E}\left[ \prod_{i \in Z'} \mathbb{I}\{N_{i,\tau^*-1} = k_i\} \mathbb{I}\{s_i\left(\overline{\mu}_{i,\tau^*-1} - \mu_i^*\right) \geq g_i^{-1}(u_i \vee g_i(0))\} \right]}_{(6)} \mathrm{d}\mathbf{u}
$$

$$(6) \leq \mathbb{P}\left[\frac{\exp\left(\sum_{i\in Z'} N_{i,\tau^*-1}\left(\frac{s_i g_i^{-1}(u_i \vee g_i(0))}{D_i}\left(\overline{\mu}_{i,\tau^*-1} - \mu_i^*\right) - \frac{\left(g_i^{-1}(u_i \vee g_i(0))\right)^2}{2D_i}\right)\right)}{\exp\left(\sum_{i\in Z'} \frac{\left(g_i^{-1}(u_i \vee g_i(0))\right)^2 k_i}{2D_i}\right)} \geq 1, (N_{i,\tau^*-1})_{i\in Z'} = \mathbf{k}\right]$$

$$\leq \mathbb{P}\left[\frac{\exp\left(\sum_{i\in Z'} N_{i,\tau^*-1}\left(\frac{s_i g_i^{-1}(u_i \vee g_i(0))}{D_i}\left(\overline{\mu}_{i,\tau^*-1} - \mu_i^*\right) - \frac{\left(g_i^{-1}(u_i \vee g_i(0))\right)^2}{2D_i}\right)\right)}{\exp\left(\sum_{i\in Z'} \frac{\left(g_i^{-1}(u_i \vee g_i(0))\right)^2 k_i}{2D_i}\right)} \geq 1\right]$$

$$\leq \frac{\mathbb{E}\left[\exp\left(\sum_{i\in Z'} N_{i,\tau^*-1}\left(\frac{s_i g_i^{-1}(u_i \vee g_i(0))}{D_i}\left(\overline{\mu}_{i,\tau^*-1} - \mu_i^*\right) - \frac{\left(g_i^{-1}(u_i \vee g_i(0))\right)^2}{2D_i}\right)\right)\right]}{\exp\left(\sum_{i\in Z'} \frac{\left(g_i^{-1}(u_i \vee g_i(0))\right)^2 k_i}{2D_i}\right)}$$

$$= \frac{\mathbb{E}\left[\exp\left(\sum_{t=1}^{\tau^*-1}\sum_{i\in Z'\cap A_t}\left(\frac{s_i g_i^{-1}(u_i \vee g_i(0))}{D_i}\left(X_{i,t} - \mu_i^*\right) - \frac{\left(g_i^{-1}(u_i \vee g_i(0))\right)^2}{2D_i}\right)\right)\right]}{\exp\left(\sum_{i\in Z'} \frac{\left(g_i^{-1}(u_i \vee g_i(0))\right)^2 k_i}{2D_i}\right)}.$$

From Assumption 4, we have that

$$M_\tau = \exp\left(\sum_{t=1}^{\tau-1}\sum_{i\in Z'\cap A_t}\left(\frac{s_i g_i^{-1}(u_i \vee g_i(0))}{D_i}(X_{i,t} - \mu_i^*) - \frac{\left(g_i^{-1}(u_i \vee g_i(0))\right)^2}{2D_i}\right)\right)$$

is a supermartingale:

$$\mathbb{E}[M_\tau | \mathcal{F}_{\tau-1}] = M_{\tau-1}\mathbb{E}\left[\exp\left(\sum_{i\in Z'\cap A_{\tau-1}}\left(\frac{s_i g_i^{-1}(u_i \vee g_i(0))}{D_i}(X_{i,\tau-1} - \mu_i^*) - \frac{\left(g_i^{-1}(u_i \vee g_i(0))\right)^2}{2D_i}\right)\right)\Bigg| \mathcal{F}_{\tau-1}\right]$$

$$\leq M_{\tau-1}.$$

Since $\tau^*$ is a stopping time with respect to $\mathcal{F}_\tau$, we have from Doob's optional sampling theorem for non-negative supermartingales[6] that $\mathbb{E}[M_{\tau^*}] \leq 1$. Therefore,

$$(6) \leq \exp\left(-\sum_{i\in Z'} \frac{\left(g_i^{-1}(u_i \vee g_i(0))\right)^2 k_i}{2D_i}\right).$$

Now, we want to use the following fact (see Chang et al. [2011]): if $\eta \sim \mathcal{N}(0,1)$, then with $\beta > 1$,

$$\sqrt{\frac{2e}{\pi}}\frac{\sqrt{\beta-1}}{\beta}e^{-\beta x^2/2} \leq \mathbb{P}[|\eta| \geq x].$$

Indeed, this gives

$$\sqrt{\frac{2e}{\pi}}\frac{\sqrt{\beta-1}}{\beta}\exp\left(-\frac{\left(g_i^{-1}(u_i \vee g_i(0))\right)^2 k_i}{2D_i}\right) \leq \mathbb{P}\left[|\eta_i| \geq g_i^{-1}(u_i \vee g_i(0))\sqrt{\frac{k_i}{\beta D_i}}\right],$$

where $\boldsymbol{\eta} \sim \mathcal{N}(0,1)^{\otimes Z'}$. Thus,

$$(5) \leq \left(\sqrt{\frac{\pi}{2e}}\frac{2\beta}{\sqrt{\beta-1}}\right)^{|Z'|}\int_{\mathbf{u}\in\mathbb{R}_+^{Z'}}\prod_{i\in Z'}\mathbb{P}\left[\sqrt{\frac{\beta D_i}{k_i}}|\eta_i| \geq g_i^{-1}(u_i \vee g_i(0))\right]\mathrm{d}\mathbf{u}$$

$$= \left(\sqrt{\frac{\pi}{2e}}\frac{2\beta}{\sqrt{\beta-1}}\right)^{|Z'|}\int_{\mathbf{u}\in\mathbb{R}_+^{Z'}}\prod_{i\in Z'}\mathbb{P}\left[g_i\left(\sqrt{\frac{\beta D_i}{k_i}}|\eta_i|\right) \geq u_i \vee g_i(0)\right]\mathrm{d}\mathbf{u}$$

$$= \left( \sqrt{\frac{\pi}{2e}} \frac{2\beta}{\sqrt{\beta - 1}} \right)^{|Z'|} \int_{\mathbf{u} \in \mathbb{R}_+^{Z'}} \prod_{i \in Z'} \mathbb{P}\left[ g_i\left( \sqrt{\frac{\beta D_i}{k_i}} |\eta_i| \right) \geq u_i \right] \mathrm{d}\mathbf{u}$$

$$= \left( \sqrt{\frac{\pi}{2e}} \frac{2\beta}{\sqrt{\beta - 1}} \right)^{|Z'|} \prod_{i \in Z'} \int_0^\infty \mathbb{P}\left[ g_i\left( \sqrt{\frac{\beta D_i}{k_i}} |\eta_i| \right) \geq u_i \right] \mathrm{d}u_i$$

$$= \left( \sqrt{\frac{\pi}{2e}} \frac{2\beta}{\sqrt{\beta - 1}} \right)^{|Z'|} \prod_{i \in Z'} \mathbb{E}\left[ g_i\left( \sqrt{\frac{\beta D_i}{k_i}} |\eta_i| \right) \right].$$

We now want to bound $\mathbb{E}\left[ g_i\left( \sqrt{\frac{\beta D_i}{k_i}} |\eta_i| \right) \right]$. We define $\alpha = 2 - \sqrt{2}$, the unique solution in $(1/2, 1)$ of $\alpha - 1/2 = (\alpha - 1)^2/2$. Notice that $\alpha - 1/2 \geq 1/12$. Define $\varepsilon_i \triangleq \varepsilon\sqrt{\frac{k_i}{\beta D_i}}$. By definition, we have

$$\mathbb{E}\left[ g_i\left( \sqrt{\frac{\beta D_i}{k_i}} |\eta_i| \right) \right] = \int_{-\infty}^{+\infty} \frac{e^{-x^2/2}}{\int_{x-\varepsilon_i}^{x+\varepsilon_i} e^{-y^2/2}\mathrm{d}y} \mathrm{d}x - 1$$

$$= \underbrace{2\int_{\alpha\varepsilon_i}^{+\infty} \frac{1}{\int_{x-\varepsilon_i}^{x+\varepsilon_i} e^{-\frac{y^2-x^2}{2}}\mathrm{d}y} \mathrm{d}x}_{A_1} + \underbrace{\int_{-\alpha\varepsilon_i}^{\alpha\varepsilon_i} \frac{e^{-x^2/2}}{\int_{x-\varepsilon_i}^{x+\varepsilon_i} e^{-y^2/2}\mathrm{d}y} \mathrm{d}x - 1}_{A_2}.$$

We first bound $A_1$. With the change of variable $u = y - x$, we get:

$$A_1 = 2\int_{\alpha\varepsilon_i}^{+\infty} \frac{1}{\int_{-\varepsilon_i}^{\varepsilon_i} e^{-u^2/2 - ux}\mathrm{d}u} \mathrm{d}x$$

$$\leq 2\int_{\alpha\varepsilon_i}^{+\infty} \frac{1}{\int_{-\varepsilon_i}^{0} e^{-u^2/2 - ux}\mathrm{d}u} \mathrm{d}x$$

Note that for $x \geq \alpha\varepsilon_i$ and $u \in [-\varepsilon_i, 0]$, $-u^2/2 - ux \geq -(1 - \frac{1}{2\alpha})ux$ and thus:

$$A_1 \leq 2\int_{\alpha\varepsilon_i}^{+\infty} \frac{1}{\int_{-\varepsilon_i}^{0} e^{-(1-\frac{1}{2\alpha})ux}\mathrm{d}u} \mathrm{d}x$$

$$= 2\int_{\alpha\varepsilon_i}^{+\infty} \frac{(1 - \frac{1}{2\alpha})x}{e^{(1-\frac{1}{2\alpha})\varepsilon_i x} - 1} \mathrm{d}x. \tag{7}$$

We distinguish two regimes. First, if $\varepsilon_i^2 \geq 12$, then

$$(7) \leq \frac{2e^{(\alpha-\frac{1}{2})\varepsilon_i^2}}{e^{(\alpha-\frac{1}{2})\varepsilon_i^2} - 1} \int_{\alpha\varepsilon_i}^{+\infty} \left( 1 - \frac{1}{2\alpha} \right) xe^{-(1-\frac{1}{2\alpha})\varepsilon_i x}\mathrm{d}x$$

$$= \frac{2e^{(\alpha-\frac{1}{2})\varepsilon_i^2}}{e^{(\alpha-\frac{1}{2})\varepsilon_i^2} - 1} \frac{1}{(1 - \frac{1}{2\alpha})\varepsilon_i^2} \int_{(\alpha-\frac{1}{2})\varepsilon_i^2}^{+\infty} xe^{-x}\mathrm{d}x$$

$$= \frac{2e^{(\alpha-\frac{1}{2})\varepsilon_i^2}}{e^{(\alpha-\frac{1}{2})\varepsilon_i^2} - 1} \frac{1}{(1 - \frac{1}{2\alpha})\varepsilon_i^2} \left[ -(x+1)e^{-x} \right]_{(\alpha-\frac{1}{2})\varepsilon_i^2}^{\infty}$$

$$= \frac{2e^{(\alpha-\frac{1}{2})\varepsilon_i^2}}{e^{(\alpha-\frac{1}{2})\varepsilon_i^2} - 1} \frac{1}{(1 - \frac{1}{2\alpha})\varepsilon_i^2} \left( \left( \alpha - \frac{1}{2} \right)\varepsilon_i^2 + 1 \right) e^{-(\alpha-\frac{1}{2})\varepsilon_i^2}$$

$$= \frac{2}{e^{(\alpha-\frac{1}{2})\varepsilon_i^2} - 1} \left( \alpha + \frac{\alpha}{(\alpha - \frac{1}{2})\varepsilon_i^2} \right)$$

$$\leq 4e^{-\varepsilon_i^2/12}.$$

Otherwise, we have

$$(7) = \frac{2(1 - \frac{1}{2\alpha})}{\varepsilon_i^2} \int_{\alpha\varepsilon_i^2}^{\infty} \frac{u}{e^{(1-\frac{1}{2\alpha})u} - 1} \mathrm{d}u$$

$$\leq \frac{2(1 - \frac{1}{2\alpha})}{\varepsilon_i^2} \int_0^\infty \frac{u}{e^{(1-\frac{1}{2\alpha})u} - 1} \mathrm{d}u$$

$$= \frac{2(1 - \frac{1}{2\alpha})}{\varepsilon_i^2} \frac{\pi^2}{6\left(1 - \frac{1}{2\alpha}\right)^2}$$

$$\leq \frac{24\beta D_i}{\varepsilon^2}.$$

We now bound $A_2$. As $x \in [-\alpha\varepsilon_i, \alpha\varepsilon_i]$, it comes that $[-(1-\alpha)\varepsilon_i, (1-\alpha)\varepsilon_i] \subset [x - \varepsilon_i, x + \varepsilon_i]$. This implies that

$$A_2 \leq \frac{\int_{-\alpha\varepsilon_i}^{\alpha\varepsilon_i} e^{-x^2/2}\mathrm{d}x}{\int_{-(1-\alpha)\varepsilon_i}^{(1-\alpha)\varepsilon_i} e^{-x^2/2}\mathrm{d}x} - 1$$

$$= \frac{2\int_{(1-\alpha)\varepsilon_i}^{\alpha\varepsilon_i} e^{-x^2/2}\mathrm{d}x}{\int_{-(1-\alpha)\varepsilon_i}^{(1-\alpha)\varepsilon_i} e^{-x^2/2}\mathrm{d}x}$$

$$\leq \frac{2\int_{(1-\alpha)\varepsilon_i}^{\infty} e^{-x^2/2}\mathrm{d}x}{\int_{-(1-\alpha)\varepsilon_i}^{(1-\alpha)\varepsilon_i} e^{-x^2/2}\mathrm{d}x}$$

$$\leq \frac{e^{-(1-\alpha)^2\varepsilon_i^2/2}}{1 - e^{-(1-\alpha)^2\varepsilon_i^2/2}} \leq \left(1 + \frac{12}{\varepsilon_i^2}\right)e^{-\varepsilon_i^2/12}.$$

The penultimate inequality relies on $\int_x^\infty e^{-u^2/2}\mathrm{d}u \leq \sqrt{\frac{\pi}{2}}e^{-x^2/2}$ (see Jacobs and Wozencraft [1965], eq. (2.122)). We obtain again two regimes: $2e^{-\varepsilon_i^2/12}$ if $\varepsilon_i^2 \geq 12$, and $1 + \frac{12\beta D_i}{\varepsilon^2}$ otherwise. To summarize, we proved that

$$(5) \leq \left(\sqrt{\frac{\pi}{2e}}\frac{2\beta}{\sqrt{\beta-1}}\right)^{|Z'|} \prod_{i\in Z'} \left(\mathbb{I}\left\{\varepsilon^2\frac{k_i}{\beta D_i} < 12\right\}\left(1 + 36\frac{\beta D_i}{\varepsilon^2}\right) + \mathbb{I}\left\{\varepsilon^2\frac{k_i}{\beta D_i} \geq 12\right\}6e^{-\varepsilon^2\frac{k_i}{12\beta D_i}}\right).$$

After the summation on $\mathbf{k}$, on $Z'$, on $q$, and on $Z$, we obtain that there exists two constants $C, C'$ such that

$$\sum_{Z\subset A^*, \; Z\neq\emptyset} \sum_{q\geq 1} \sum_{Z'\subset Z, \; Z'\neq\emptyset} \sum_{\mathbf{k}\in[q..\infty)^{Z'}} (5) \leq \left(C\varepsilon^{-2}\beta\max_i D_i\right)\left(\frac{C'\beta}{\sqrt{\beta-1}}\varepsilon^{-4}\beta^2\max_i D_i^2\right)^{m^*}.$$

Thus,

$$\sum_{t\in[T]} \mathbb{E}[\Delta(A_t)\mathbb{I}\{\mathfrak{M}_t \wedge \mathfrak{Z}_t \wedge \neg\mathfrak{C}_t\}] \leq \Delta_{\max}\left(C\varepsilon^{-2}\beta\max_i D_i\right)\left(\frac{C'\beta}{\sqrt{\beta-1}}\varepsilon^{-4}\beta^2\max_i D_i^2\right)^{m^*}.$$

# D    Proof of Theorem 3 (CLIP CTS-GAUSSIAN for linear rewards)

In this section, we provide an analysis for the regret bound of CLIP CTS-GAUSSIAN, which is stated completely as follows.

**Theorem.** *The policy* CLIP CTS-GAUSSIAN *has regret bounded by*

$$\sum_{i\in[n]} \frac{128\left(4\log_2^2(4\sqrt{m})\beta D_i\log(2^m|\mathcal{A}|T) \wedge m\Gamma_{ii}(\log(T) + 4\log\log(T))\right)}{\Delta_{i,\min}} + \Delta_{\max}(1 + 5.2n)$$

$$+ \frac{nm^2\Delta_{\max}}{\left(\frac{\Delta_{\min}}{2B} - (m^*(m^* + 1)/2 + 1)\varepsilon\right)^2} + \Delta_{\max}\left(C\varepsilon^{-2}\beta\max_i D_i\right)\left(\frac{C'}{\sqrt{\beta-1}}\varepsilon^{-4}\beta^3\max_i D_i^2\right)^{m^*},$$

*where* $C, C'$ *are two universal constants, and* $\varepsilon \in (0,1)$ *is such that* $\Delta_{\min}/(2B) - (m^{*2} + 1)\varepsilon > 0$.

More precisely, notice that the modification on the sample $\boldsymbol{\theta}_t$ has an impact only in two places in the analysis: in the concentration bound and in the event controlling optimism. We detail these two points in the following.

## D.1 Concentration bound

In this subsection, we provide the concentration bound of CLIP CTS-GAUSSIAN. Our strategy here is to either use the concentration from $\boldsymbol{\mu}_t$ or from $\boldsymbol{\theta}_t$, depending on which regime is the best for each arm. Thus, we define $S \triangleq \left\{ i \in [n], \Gamma_{ii} m(\log(T) + 4 \log\log(T)) \geq 4 \log_2^2(4\sqrt{m})\beta D_i \log(|A|2^m T) \right\}$. We have the following lemma.

**Lemma 7.**
$$\mathbb{P}\left[ \mathbf{e}_{A_t \cap S}^\intercal \left( \overline{\boldsymbol{\mu}}_{t-1} \vee \boldsymbol{\theta}_t \wedge \boldsymbol{\mu}_t - \overline{\boldsymbol{\mu}}_{t-1} \right) \geq \sqrt{2 \log(|\mathcal{A}|2^m T) \sum_{i \in A_t \cap S} \beta D_i / N_{i,t-1}} \,\bigg|\, \mathcal{H}_t \right] \leq 1/T.$$

*Proof.* We define the functions
$$\alpha_t(A) \triangleq \sqrt{2 \log(|\mathcal{A}|2^m T) \sum_{i \in A} \frac{\beta D_i}{N_{i,t-1}}}, \quad \text{and} \quad \lambda_t(A) \triangleq \frac{\alpha_t(A)}{\sum_{i \in A} \beta D_i / N_{i,t-1}},$$
we have
$$\mathbb{P}\left[ \mathbf{e}_{A_t \cap S}^\intercal \left( \overline{\boldsymbol{\mu}}_{t-1} \vee \boldsymbol{\theta}_t \wedge \boldsymbol{\mu}_t - \overline{\boldsymbol{\mu}}_{t-1} \right) \geq \alpha_t(A_t \cap S) \big| \mathcal{H}_t \right]$$
$$\leq \sum_{A \in \mathcal{A}} \mathbb{P}\left[ \mathbf{e}_{A \cap S}^\intercal \left( \overline{\boldsymbol{\mu}}_{t-1} \vee \boldsymbol{\theta}_t - \overline{\boldsymbol{\mu}}_{t-1} \right) \geq \alpha_t(A \cap S) \big| \mathcal{H}_t \right]$$
$$\leq \sum_{A \in \mathcal{A}} e^{-\lambda_t(A \cap S)\alpha_t(A \cap S)} \mathbb{E}\left[ e^{\lambda_t(A \cap S) \left\| \mathbf{e}_{A \cap S} \odot \left( 0 \vee \left( \boldsymbol{\theta}_t - \overline{\boldsymbol{\mu}}_{t-1} \right) \right) \right\|_1} \big| \mathcal{H}_t \right]$$
$$\leq \sum_{A \in \mathcal{A}} e^{-\lambda_t(A \cap S)\alpha_t(A \cap S)} \prod_{i \in A \cap S} \mathbb{E}\left[ e^{\lambda_t(A \cap S) \left( 0 \vee \left( \theta_{i,t} - \overline{\mu}_{i,t-1} \right) \right)} \big| \mathcal{H}_t \right]$$
$$\leq \sum_{A \in \mathcal{A}} e^{-\lambda_t(A \cap S)\alpha_t(A \cap S)} \prod_{i \in A \cap S} \mathbb{E}\left[ 1 + e^{\lambda_t(A \cap S) \left( \theta_{i,t} - \overline{\mu}_{i,t-1} \right)} \big| \mathcal{H}_t \right]$$
$$\leq \sum_{A \in \mathcal{A}} e^{-\lambda_t(A \cap S)\alpha_t(A \cap S)} \prod_{i \in A \cap S} \mathbb{E}\left[ 2 e^{\lambda_t(A \cap S) \left( \theta_{i,t} - \overline{\mu}_{i,t-1} \right)} \big| \mathcal{H}_t \right]$$
$$\leq \sum_{A \in \mathcal{A}} 2^{|A \cap S|} e^{-\lambda_t(A \cap S)\alpha_t(A \cap S)} e^{\lambda_t(A \cap S)^2 \sum_{i \in A \cap S} \beta D_i / (2 N_{i,t-1})}$$
$$\leq 1/T.$$
$\square$

We now use the definition of $\boldsymbol{\mu}_t$ to have
$$\mathbf{e}_{A_t \cap S^c}^\intercal \left( \overline{\boldsymbol{\mu}}_{t-1} \vee \boldsymbol{\theta}_t \wedge \boldsymbol{\mu}_t - \overline{\boldsymbol{\mu}}_{t-1} \right) \leq \mathbf{e}_{A_t \cap S^c}^\intercal \left( \boldsymbol{\mu}_t - \overline{\boldsymbol{\mu}}_{t-1} \right) = \sum_{i \in A_t \cap S^c} \sqrt{\Gamma_{ii} \frac{2(\log(t) + 4 \log\log(t))}{N_{i,t-1}}}.$$

To conclude, we have the following event
$$\mathfrak{A}_t \triangleq \left\{ \Delta_t \leq \sqrt{8 \log(|\mathcal{A}|2^m T) \sum_{i \in A_t \cap S} \beta D_i / N_{i,t-1}} + \sum_{i \in A_t \cap S^c} \sqrt{\Gamma_{ii} \frac{8(\log(t) + 4 \log\log(t))}{N_{i,t-1}}} \right\}.$$

Using Proposition 4, we have
$$\sum_{t \in [T]} \mathbb{E}[\Delta_t \mathbb{I}\{\mathfrak{A}_t\}] \leq \sum_{t \in [T]} \mathbb{E}\left[ \Delta_t \mathbb{I}\left\{ \Delta_t \leq 2 \sqrt{8 \log(|\mathcal{A}|2^m T) \sum_{i \in A_t \cap S} \beta D_i / N_{i,t-1}} \right\} \right]$$
$$+ \sum_{t \in [T]} \mathbb{E}\left[ \Delta_t \mathbb{I}\left\{ \Delta_t \leq 2 \sum_{i \in A_t \cap S^c} \sqrt{\Gamma_{ii} \frac{8(\log(t) + 4 \log\log(t))}{N_{i,t-1}}} \right\} \right].$$

We can thus apply Theorem 5 and Theorem 4 (see Appendix E) to get the bound
$$512 \log_2^2(4\sqrt{m}) \sum_{i \in S} \Delta_{i,\min}^{-1} \beta D_i \log(|\mathcal{A}|2^m T) + 128 m \sum_{i \in S^c} \Delta_{i,\min}^{-1} \Gamma_{ii} (\log(T) + 4 \log\log(T))$$

## D.2 Optimism

In this subsection, we examine the theoretical impact of considering CLIP CTS-GAUSSIAN on the optimism-controlling event (event $\neg\mathfrak{C}_t$), in the case of linear rewards. For this purpose, we modify the beginning of Step 4 in the analysis by considering the following events.

- $\mathfrak{Z}_t \triangleq \{\Delta_t > 0\}$
- $\mathfrak{C}_t \triangleq \left\{ \mathbf{e}_{A_t}^\top \widetilde{\boldsymbol{\theta}}_t > \mathbf{e}_{A^*}^\top \boldsymbol{\mu}^* - (m^*(m^*+1)/2+1)\varepsilon \right\}$
- $\mathfrak{R}(\boldsymbol{\theta}', Z) \triangleq \left\{ \forall A \in \arg\max_{A' \in \mathcal{A}} \mathbf{e}_{A'}^\top(\boldsymbol{\theta}') \text{ we have } Z \subset A, \ \mathbf{e}_{\mathrm{Oracle}(\boldsymbol{\theta}')}^\top \boldsymbol{\theta}' > \mathbf{e}_{A^*}^\top \boldsymbol{\mu}^* - (m^*(m^*+1)/2+1)\varepsilon \right\}$
- $\mathfrak{S}_t(Z) \triangleq \left\{ \forall \boldsymbol{\theta}' \text{ s.t. } 0 \leq (\boldsymbol{\mu}^* - \boldsymbol{\theta}') \odot \mathbf{e}_Z \leq \varepsilon \mathbf{e}_Z, \ \mathfrak{R}(\boldsymbol{\theta}' \odot \mathbf{e}_Z + \widetilde{\boldsymbol{\theta}}_t \odot \mathbf{e}_{Z^c}, Z) \text{ holds} \right\}$
- $\mathfrak{T}_t(Z) \triangleq \left\{ \exists i \in Z, \ \mu_i^* - \mu_i^* \wedge \widetilde{\theta}_{i,t} > \varepsilon \right\}$.
- $\mathfrak{J}_t \triangleq \{\forall i \in [n], \mu_i^* \leq \mu_{i,t}\}$

In the above events, $\widetilde{\boldsymbol{\theta}}_t$ is $\boldsymbol{\mu}_t \wedge \boldsymbol{\theta}_t \vee \overline{\boldsymbol{\mu}}_t$. The last event $\mathfrak{J}_t$ holds with probability at least $1 - n/(t\log^2(t))$ from Hoeffding's inequality [Hoeffding, 1963]. We thus assume that this event hods in the following, since the regret under the complementary event is bounded by $3.2 n\Delta_{\max}$. We first state the following lemma.

**Lemma 8.**
$$\mathfrak{Z}_t, \neg\mathfrak{C}_t \Rightarrow \exists Z \subset A^*, \ Z \neq \emptyset \text{ s.t. the event } \mathfrak{S}_t(Z) \wedge \mathfrak{T}_t(Z) \text{ holds.}$$

This allows us to consider the success probability $\mathbb{P}[\neg\mathfrak{T}_t(Z)|\mathcal{H}_t]$ in the analysis. Notice however that $Z \subset \mathrm{Oracle}\left(\left(\boldsymbol{\mu}^* \wedge \widetilde{\boldsymbol{\theta}}_t\right) \odot \mathbf{e}_Z + \widetilde{\boldsymbol{\theta}}_t \odot \mathbf{e}_{Z^c}\right)$, that is guaranteed when $\mathfrak{S}_t(Z) \wedge \neg\mathfrak{T}_t(Z)$ holds, does not necessarily implies that $Z \subset \mathrm{Oracle}\left(\widetilde{\boldsymbol{\theta}}_t\right)$. However, it turns out that we have $Z \subset A$ for all $A \in \arg\max_{A' \in \mathcal{A}} \mathbf{e}_{A'}^\top\left(\left(\boldsymbol{\mu}^* \wedge \widetilde{\boldsymbol{\theta}}_t\right) \odot \mathbf{e}_Z + \widetilde{\boldsymbol{\theta}}_t \odot \mathbf{e}_{Z^c}\right)$ implies that $Z \subset A$ for all $A \in \arg\max_{A' \in \mathcal{A}} \mathbf{e}_{A'}^\top\left(\widetilde{\boldsymbol{\theta}}_t\right)$. This last fact is from Lemma 9, with $\boldsymbol{\eta} = \left(\boldsymbol{\mu}^* \wedge \widetilde{\boldsymbol{\theta}}_t\right) \odot \mathbf{e}_Z + \widetilde{\boldsymbol{\theta}}_t \odot \mathbf{e}_{Z^c}$ and $\boldsymbol{\delta} = \left(\widetilde{\boldsymbol{\theta}}_t - \boldsymbol{\mu}^* \wedge \widetilde{\boldsymbol{\theta}}_t\right) \odot \mathbf{e}_Z$.

**Lemma 9.** Let $\boldsymbol{\eta} \in \mathbb{R}^n$, $\boldsymbol{\delta} \in \mathbb{R}_+^n$ such that for all $A \in \arg\max_{A' \in \mathcal{A}} \mathbf{e}_{A'}^\top \boldsymbol{\eta}$, we have $Z \subset A$. Then, for all $A \in \arg\max_{A' \in \mathcal{A}} \mathbf{e}_{A'}^\top(\boldsymbol{\eta} + \boldsymbol{\delta} \odot \mathbf{e}_Z)$, we have $Z \subset A$.

It now remains to explain how to handle the probability $\mathbb{P}[\neg\mathfrak{T}_t(Z)|\mathcal{H}_t]$ in the analysis. Notice that from the high probability event $\mathfrak{J}_t$, it suffices to treat the case $\widetilde{\boldsymbol{\theta}}_t = \boldsymbol{\theta}_t \vee \overline{\boldsymbol{\mu}}_t$. We provide here the places where the analysis differs, the rest of the proof remains unchanged.

- We use that $\mathbb{P}[\neg\mathfrak{T}_t(Z)|\mathcal{H}_t] = \mathbb{P}\left[\forall i \in Z, \ \varepsilon \vee \left(\mu_i^* - \overline{\mu}_{i,t-1}\right) - 0 \vee \left(\theta_{i,t} - \overline{\mu}_{i,t-1}\right) \leq \varepsilon \middle| \mathcal{H}_t\right]$, is a product of functions that are decreasing with respect to $\varepsilon \vee \left(\mu_i^* - \overline{\mu}_{i,t-1}\right)$.
- We use that $\varepsilon \vee \left(\mu_i^* - \overline{\mu}_{i,t-1}\right) \geq g_i^{-1}(u_i \vee g_i(\varepsilon))$ is equivalent to $\mu_i^* - \overline{\mu}_{i,t-1} \geq g_i^{-1}(u_i \vee g_i(\varepsilon))$. Thus, we don't sum on $\mathbf{s}$, and can use Assumption 4 with $\boldsymbol{\lambda} \in \mathbb{R}_+^n$.

*Proof of Lemma 8.* It is sufficient to prove that

$$\mathfrak{Z}_t, \neg\mathfrak{C}_t \Rightarrow \exists Z \subset A^*, \ Z \neq \emptyset \text{ s.t. } \mathfrak{S}_t(Z) \text{ holds,} \tag{8}$$

because $\neg\mathfrak{C}_t$ and $\mathfrak{S}_t(Z)$ together imply $\mathfrak{T}_t(Z)$. Indeed, see that from $\neg\mathfrak{T}_t(Z)$, we can plug $\boldsymbol{\theta}' = \boldsymbol{\mu}^* \wedge \widetilde{\boldsymbol{\theta}}_t$ into $\mathfrak{S}_t(Z)$ to get

$$\mathbf{e}_{A_t}^\top \widetilde{\boldsymbol{\theta}}_t = \max_{A \in \mathcal{A}} \mathbf{e}_A^\top \widetilde{\boldsymbol{\theta}}_t$$
$$\geq \max_{A \in \mathcal{A}} \mathbf{e}_A^\top \left(\boldsymbol{\theta}' \odot \mathbf{e}_Z + \widetilde{\boldsymbol{\theta}}_t \odot \mathbf{e}_{Z^c}\right)$$

$$= \mathbf{e}_{\mathrm{Oracle}\left(\boldsymbol{\theta}' \odot \mathbf{e}_Z + \widetilde{\boldsymbol{\theta}}_t \odot \mathbf{e}_{Z^c}\right)}^{\mathsf{T}} \left(\boldsymbol{\theta}' \odot \mathbf{e}_Z + \widetilde{\boldsymbol{\theta}}_t \odot \mathbf{e}_{Z^c}\right)$$
$$> \mathbf{e}_{A^*}^{\mathsf{T}} \boldsymbol{\mu}^* - (m^*(m^* + 1)/2 + 1)\varepsilon,$$

giving $\mathfrak{C}_t$. To prove (8), we first consider the choice $Z = Z_1 = A^*$. Two cases can be distinguished:

1a) $\forall \boldsymbol{\theta}'$ s.t. $0 \le (\boldsymbol{\mu}^* - \boldsymbol{\theta}') \odot \mathbf{e}_{A^*} \le \varepsilon \mathbf{e}_{A^*}$, we have $A^* \subset A$ for any action $A \in \arg\max_{A' \in \mathcal{A}} \mathbf{e}_{A'}^{\mathsf{T}} \left(\boldsymbol{\theta}' \odot \mathbf{e}_{A^*} + \widetilde{\boldsymbol{\theta}}_t \odot \mathbf{e}_{A^{*c}}\right)$.

1b) $\exists \boldsymbol{\theta}'$ s.t. $0 \le (\boldsymbol{\mu}^* - \boldsymbol{\theta}') \odot \mathbf{e}_{A^*} \le \varepsilon \mathbf{e}_{A^*}$ such that $A^* \not\subset A$ for some action $A \in \arg\max_{A' \in \mathcal{A}} \mathbf{e}_{A'}^{\mathsf{T}} \left(\boldsymbol{\theta}' \odot \mathbf{e}_{A^*} + \widetilde{\boldsymbol{\theta}}_t \odot \mathbf{e}_{A^{*c}}\right)$.

**1a)** For the first case, consider any vector $\boldsymbol{\theta}'$ such that $0 \le (\boldsymbol{\mu}^* - \boldsymbol{\theta}') \odot \mathbf{e}_{A^*} \overset{(9)}{\le} \varepsilon \mathbf{e}_{A^*}$ and let $A \overset{(10)}{=} \mathrm{Oracle}\left(\boldsymbol{\theta}' \odot \mathbf{e}_{A^*} + \widetilde{\boldsymbol{\theta}}_t \odot \mathbf{e}_{A^{*c}}\right)$. We can write

$$\mathbf{e}_A^{\mathsf{T}} \left(\boldsymbol{\theta}' \odot \mathbf{e}_{A^*} + \widetilde{\boldsymbol{\theta}}_t \odot \mathbf{e}_{A^{*c}}\right) \overset{(11)}{\ge} \mathbf{e}_{A^*}^{\mathsf{T}} \left(\boldsymbol{\theta}' \odot \mathbf{e}_{A^*} + \widetilde{\boldsymbol{\theta}}_t \odot \mathbf{e}_{A^{*c}}\right) \overset{(12)}{\ge} \mathbf{e}_{A^*}^{\mathsf{T}} \boldsymbol{\mu}^* - m^* \varepsilon,$$

where (11) is from (10), and (12) is from (9). This rewrites as

$$\mathbf{e}_A^{\mathsf{T}} \left(\boldsymbol{\theta}' \odot \mathbf{e}_{A^*} + \widetilde{\boldsymbol{\theta}}_t \odot \mathbf{e}_{A^{*c}}\right) \ge \mathbf{e}_{A^*}^{\mathsf{T}} \boldsymbol{\mu}^* - m^* \varepsilon > \mathbf{e}_{A^*}^{\mathsf{T}} \boldsymbol{\mu}^* - (m^*(m^* + 1)/2 + 1)\varepsilon,$$

so $\mathfrak{R}_t(\boldsymbol{\theta}' \odot \mathbf{e}_{A^*} + \widetilde{\boldsymbol{\theta}}_t \odot \mathbf{e}_{A^{*c}}, A^*)$ holds. Therefore, we have proved that $\mathfrak{S}_t(A^*)$ holds.

**1b)** For the second case, we have some vector $\boldsymbol{\theta}'$ such that $0 \overset{(13)}{\le} (\boldsymbol{\mu}^* - \boldsymbol{\theta}') \odot \mathbf{e}_{A^*} \overset{(14)}{\le} \varepsilon \mathbf{e}_{A^*}$, and some action $A \in \arg\max_{A' \in \mathcal{A}} \mathbf{e}_{A'}^{\mathsf{T}} \left(\boldsymbol{\theta}' \odot \mathbf{e}_{A^*} + \widetilde{\boldsymbol{\theta}}_t \odot \mathbf{e}_{A^{*c}}\right)$ such that $A^* \not\subset A$. We consider $Z_2 = A^* \cap A$. We first prove that $Z_2 \ne \emptyset$ by showing that if an action $S'$ is such that $S' \cap A^* \overset{(15)}{=} \emptyset$, then $A \ne S'$:

$$\mathbf{e}_{S'}^{\mathsf{T}} \left(\boldsymbol{\theta}' \odot \mathbf{e}_{A^*} + \widetilde{\boldsymbol{\theta}}_t \odot \mathbf{e}_{A^{*c}}\right) \overset{(16)}{=} \mathbf{e}_{S'}^{\mathsf{T}} \widetilde{\boldsymbol{\theta}}_t \overset{(17)}{\le} \mathbf{e}_{A_t}^{\mathsf{T}} \widetilde{\boldsymbol{\theta}}_t$$
$$\overset{(18)}{\le} \mathbf{e}_{A^*}^{\mathsf{T}} \boldsymbol{\mu}^* - (m^*(m^* + 1)/2 + 1)\varepsilon$$
$$< \mathbf{e}_{A^*}^{\mathsf{T}} \boldsymbol{\mu}^* - m^* \varepsilon$$
$$\overset{(19)}{\le} \mathbf{e}_{A^*}^{\mathsf{T}} \left(\boldsymbol{\theta}' \odot \mathbf{e}_{A^*} + \widetilde{\boldsymbol{\theta}}_t \odot \mathbf{e}_{A^{*c}}\right),$$

where (16) is from (15), (17) is from the definition of $A_t$, (18) is from $\neg \mathfrak{C}_t$ and (19) is from (14). Now, we again distinguish two cases:

2a) $\forall \boldsymbol{\theta}''$ s.t. $0 \le (\boldsymbol{\mu}^* - \boldsymbol{\theta}'') \odot \mathbf{e}_{Z_2} \le \varepsilon \mathbf{e}_{Z_2}$, we have $Z_2 \subset B$ for any action $B \in \arg\max_{A' \in \mathcal{A}} \mathbf{e}_{A'}^{\mathsf{T}} \left(\boldsymbol{\theta}'' \odot \mathbf{e}_{Z_2} + \widetilde{\boldsymbol{\theta}}_t \odot \mathbf{e}_{Z_2^c}\right)$.

2b) $\exists \boldsymbol{\theta}''$ s.t. $0 \le (\boldsymbol{\mu}^* - \boldsymbol{\theta}'') \odot \mathbf{e}_{Z_2} \le \varepsilon \mathbf{e}_{Z_2}$ such that $Z_2 \not\subset B$ for some action $B \in \arg\max_{A' \in \mathcal{A}} \mathbf{e}_{A'}^{\mathsf{T}} \left(\boldsymbol{\theta}'' \odot \mathbf{e}_{Z_2} + \widetilde{\boldsymbol{\theta}}_t \odot \mathbf{e}_{Z_2^c}\right)$.

Notice that when $0 \le (\boldsymbol{\mu}^* - \boldsymbol{\theta}'') \odot \mathbf{e}_{Z_2} \overset{(20)}{\le} \varepsilon \mathbf{e}_{Z_2}$, then

$$\mathbf{e}_A^{\mathsf{T}} \left(\boldsymbol{\theta}'' \odot \mathbf{e}_{Z_2} + \widetilde{\boldsymbol{\theta}}_t \odot \mathbf{e}_{Z_2^c}\right) \ge \mathbf{e}_A^{\mathsf{T}} \left(\boldsymbol{\theta}' \odot \mathbf{e}_{A^*} + \widetilde{\boldsymbol{\theta}}_t \odot \mathbf{e}_{A^{*c}}\right) - (m^* - 1)\varepsilon. \qquad (21)$$

Indeed, (21) is a consequence of

$$\mathbf{e}_A^{\mathsf{T}} \left(\boldsymbol{\theta}'' \odot \mathbf{e}_{Z_2} + \widetilde{\boldsymbol{\theta}}_t \odot \mathbf{e}_{Z_2^c} - \boldsymbol{\theta}' \odot \mathbf{e}_{A^*} - \widetilde{\boldsymbol{\theta}}_t \odot \mathbf{e}_{A^{*c}}\right) = \mathbf{e}_{Z_2}^{\mathsf{T}} (\boldsymbol{\theta}'' - \boldsymbol{\theta}')$$

$$= \mathbf{e}_{Z_2}^{\mathsf{T}}(\boldsymbol{\theta}'' - \boldsymbol{\mu}^*) + \mathbf{e}_{Z_2}^{\mathsf{T}}(\boldsymbol{\mu}^* - \boldsymbol{\theta}')$$
$$\geq -\varepsilon(m^* - 1) + 0,$$

where we used (20), (13) and that $Z_2$ is strictly included in $A^*$.

**2a)** For the first case, considering any vector $\boldsymbol{\theta}''$ such that $0 \leq (\boldsymbol{\mu}^* - \boldsymbol{\theta}'') \odot \mathbf{e}_{Z_2} \leq \varepsilon \mathbf{e}_{Z_2}$, we have with $B = \mathrm{Oracle}\Big(\boldsymbol{\theta}'' \odot \mathbf{e}_{Z_2} + \widetilde{\boldsymbol{\theta}}_t \odot \mathbf{e}_{Z_2{}^c}\Big)$ that

$$\mathbf{e}_B^{\mathsf{T}}\Big(\boldsymbol{\theta}'' \odot \mathbf{e}_{Z_2} + \widetilde{\boldsymbol{\theta}}_t \odot \mathbf{e}_{Z_2{}^c}\Big) \geq \mathbf{e}_A^{\mathsf{T}}\Big(\boldsymbol{\theta}'' \odot \mathbf{e}_{Z_2} + \widetilde{\boldsymbol{\theta}}_t \odot \mathbf{e}_{Z_2{}^c}\Big)$$
$$\overset{(22)}{\geq} \mathbf{e}_A^{\mathsf{T}}\Big(\boldsymbol{\theta}' \odot \mathbf{e}_{A^*} + \widetilde{\boldsymbol{\theta}}_t \odot \mathbf{e}_{A^*{}^c}\Big) - (m^* - 1)\varepsilon$$
$$\geq \mathbf{e}_{A^*}^{\mathsf{T}}\Big(\boldsymbol{\theta}' \odot \mathbf{e}_{A^*} + \widetilde{\boldsymbol{\theta}}_t \odot \mathbf{e}_{A^*{}^c}\Big) - (m^* - 1)\varepsilon$$
$$\overset{(23)}{\geq} \mathbf{e}_{A^*}^{\mathsf{T}} \boldsymbol{\mu}^* - m^* \varepsilon - (m^* - 1)\varepsilon,$$

where (22) uses (21) and (23) uses (14). This rewrites as

$$\mathbf{e}_B^{\mathsf{T}}\Big(\boldsymbol{\theta}'' \odot \mathbf{e}_{Z_2} + \widetilde{\boldsymbol{\theta}}_t \odot \mathbf{e}_{Z_2{}^c}\Big) \geq \mathbf{e}_{A^*}^{\mathsf{T}} \boldsymbol{\mu}^* - (m^*(m^* + 1)/2 + 1)\varepsilon,$$

so $\mathfrak{R}_t(\boldsymbol{\theta}' \odot \mathbf{e}_{Z_2} + \widetilde{\boldsymbol{\theta}}_t \odot \mathbf{e}_{Z_2{}^c}, Z_2)$ holds, and thus we proved that $\mathfrak{S}_t(Z_2)$ holds.

**2b)** For the second case, we have a vector $\boldsymbol{\theta}''$ such that $0 \leq (\boldsymbol{\mu}^* - \boldsymbol{\theta}'') \odot \mathbf{e}_{Z_2} \leq \varepsilon \mathbf{e}_{Z_2}$ and an action $B \in \arg\max_{A' \in \mathcal{A}} \mathbf{e}_{A'}^{\mathsf{T}}\Big(\boldsymbol{\theta}'' \odot \mathbf{e}_{Z_2} + \widetilde{\boldsymbol{\theta}}_t \odot \mathbf{e}_{Z_2{}^c}\Big)$ such that $Z_2 \not\subset B$. We consider $Z_3 = Z_2 \cap B$. Again, $Z_3 \neq \emptyset$ because for any $S'$ such that $S' \cap Z_2 = \emptyset$, we have $S' \neq \mathrm{Oracle}\Big(\boldsymbol{\theta}'' \odot \mathbf{e}_{Z_2} + \widetilde{\boldsymbol{\theta}}_t \odot \mathbf{e}_{Z_2{}^c}\Big)$:

$$\mathbf{e}_{S'}^{\mathsf{T}}\Big(\boldsymbol{\theta}'' \odot \mathbf{e}_{Z_2} + \widetilde{\boldsymbol{\theta}}_t \odot \mathbf{e}_{Z_2{}^c}\Big) = \mathbf{e}_{S'}^{\mathsf{T}} \widetilde{\boldsymbol{\theta}}_t \leq \mathbf{e}_{A_t}^{\mathsf{T}} \widetilde{\boldsymbol{\theta}}_t$$
$$\leq \mathbf{e}_{A^*}^{\mathsf{T}} \boldsymbol{\mu}^* - (m^*(m^* + 1)/2 + 1)\varepsilon$$
$$< \mathbf{e}_{A^*}^{\mathsf{T}} \boldsymbol{\mu}^* - (m^* + (m^* - 1))\varepsilon$$
$$\leq \mathbf{e}_A^{\mathsf{T}}\Big(\boldsymbol{\theta}'' \odot \mathbf{e}_{Z_2} + \widetilde{\boldsymbol{\theta}}_t \odot \mathbf{e}_{Z_2{}^c}\Big),$$

where the last inequality is obtained in the same way as in inequalities from (22) to (23).

We could repeat the above argument and each time the size $Z_i$ is decreased by at least 1. Thus, after at most $m^* - 1$ steps, since $m^* + (m^* - 1) + (m^* - 2) + \cdots + 1 = m^*(m^* + 1)/2$ is still less than $m^*(m^* + 1)^2/2 + 1$, we could reach the end and find a $Z_i \neq \emptyset$ such that $\mathfrak{S}_t(Z_i)$ holds. $\qquad\square$

*Proof of Lemma 9.* Let's prove that $\arg\max_{A' \in \mathcal{A}} \mathbf{e}_{A'}^{\mathsf{T}}(\boldsymbol{\eta} + \boldsymbol{\delta} \odot \mathbf{e}_Z) \subset \arg\max_{A' \in \mathcal{A}} \mathbf{e}_{A'}^{\mathsf{T}} \boldsymbol{\eta}$. Consider any action $A \in \arg\max_{A' \in \mathcal{A}} \mathbf{e}_{A'}^{\mathsf{T}}(\boldsymbol{\eta} + \boldsymbol{\delta} \odot \mathbf{e}_Z)$. If $A \notin \arg\max_{A' \in \mathcal{A}} \mathbf{e}_{A'}^{\mathsf{T}} \boldsymbol{\eta}$, then there exists $B \in \arg\max_{A' \in \mathcal{A}} \mathbf{e}_{A'}^{\mathsf{T}} \boldsymbol{\eta}$ such that

$$\mathbf{e}_A^{\mathsf{T}} \boldsymbol{\eta} < \mathbf{e}_B^{\mathsf{T}} \boldsymbol{\eta}.$$

Furthermore, since $Z \subset B$ and $\boldsymbol{\delta} \geq 0$, we also have

$$\mathbf{e}_A^{\mathsf{T}}(\boldsymbol{\delta} \odot \mathbf{e}_Z) \leq \mathbf{e}_B^{\mathsf{T}}(\boldsymbol{\delta} \odot \mathbf{e}_Z),$$

so we finally have

$$\mathbf{e}_A^{\mathsf{T}}(\boldsymbol{\eta} + \boldsymbol{\delta} \odot \mathbf{e}_Z) < \mathbf{e}_B^{\mathsf{T}}(\boldsymbol{\eta} + \boldsymbol{\delta} \odot \mathbf{e}_Z),$$

contradicting that $A \in \arg\max_{A' \in \mathcal{A}} \mathbf{e}_{A'}^{\mathsf{T}}(\boldsymbol{\eta} + \boldsymbol{\delta} \odot \mathbf{e}_Z)$. $\qquad\square$

# E   General CMAB results

In this section, we state general results that are useful for every regret analysis that we conducted in this paper. The main result of the section is the following theorem, inspired from the analysis of Degenne and Perchet [2016b], that gives a regret bound under the event that the gap $\Delta_t$ is controlled by a $\ell_2$ norm type error.

**Theorem 4** (Regret bound for $\ell_2$-norm error). *For all $i \in [n]$, let $\beta_{i,T} \in \mathbb{R}_+$. For $t \geq 1$, consider the event*

$$\mathfrak{A}_t \triangleq \left\{ \Delta_t \leq \left\| \sum_{i \in A_t} \frac{\beta_{i,T}^{1/2} \mathbf{e}_i}{N_{i,t-1}^{1/2}} \right\|_2 \right\}.$$

*Then,*

$$\sum_{t=1}^{T} \mathbb{I}\{\mathfrak{A}_t\}\Delta_t \leq 32 \log_2^2(4\sqrt{m}) \sum_{i \in [n]} \beta_{i,T} \Delta_{i,\min}^{-1}.$$

*Proof.* Let $t \geq 1$. We define $\Lambda_t \triangleq \left\| \sum_{i \in A_t} \beta_{i,T}^{1/2} N_{i,t-1}^{-1/2} \mathbf{e}_i \right\|_2$. We start by a simple lower bound on $\Lambda_t$, holding for any $j \in A_t$,

$$\Lambda_t \geq \left\| \frac{\beta_{j,T}^{1/2} \mathbf{e}_j}{N_{j,t}^{1/2}} \right\|_2 = \frac{\beta_{j,T}^{1/2}}{N_{j,t}^{1/2}}. \tag{24}$$

We then use the same reverse amortisation technique than in [Wang and Chen [2017]](#).

$$
\begin{aligned}
\Lambda_t &= -\Lambda_t + \left\| \sum_{i \in A_t} \frac{2\beta_{i,T}^{1/2} \mathbf{e}_i}{N_{i,t-1}^{1/2}} \right\|_2 \\
&= -\left\| \frac{\Lambda_t \mathbf{e}_{A_t}}{\|\mathbf{e}_{A_t}\|_2} \right\|_2 + \left\| \sum_{i \in A_t} \frac{2\beta_{i,T}^{1/2} \mathbf{e}_i}{N_{i,t-1}^{1/2}} \right\|_2 \\
&\leq \left\| \sum_{i \in A_t} \left( \frac{2\beta_{i,T}^{1/2}}{N_{i,t-1}^{1/2}} - \frac{\Lambda_t}{\|\mathbf{e}_{A_t}\|_2} \right)^+ \mathbf{e}_i \right\|_2 \\
&= \left\| \sum_{i \in A_t} \left( \frac{2\beta_{i,T}^{1/2}}{N_{i,t-1}^{1/2}} - \frac{\Lambda_t}{\|\mathbf{e}_{A_t}\|_2} \right)^+ \mathbb{I}\left\{ \Lambda_t \geq \frac{\beta_{i,T}^{1/2}}{N_{i,t-1}^{1/2}} \right\} \mathbf{e}_i \right\|_2 \qquad \text{Using (24)} \\
&\leq \left\| \sum_{i \in A_t} \mathbb{I}\left\{ 2\Lambda_t \geq \frac{2\beta_{i,T}^{1/2}}{N_{i,t-1}^{1/2}} \geq \frac{\Lambda_t}{\|\mathbf{e}_{A_t}\|_2} \right\} \frac{2\beta_{i,T}^{1/2} \mathbf{e}_i}{N_{i,t-1}^{1/2}} \right\|_2.
\end{aligned}
$$

We now decompose the interval $[2, 1/\|\mathbf{e}_{A_t}\|_2]$ using a peeling:

$$[2, 1/\|\mathbf{e}_{A_t}\|_2] \subset \bigcup_{k=0}^{\lceil \log_2(\|\mathbf{e}_{A_t}\|_2) \rceil} [2^{-k}, 2^{1-k}].$$

This induces a partition of the set of indices:

$$\mathbb{I}\left\{ i \in A_t, \ 2\Lambda_t \geq \frac{2\beta_{i,T}^{1/2}}{N_{i,t-1}^{1/2}} \geq \frac{\Lambda_t}{\|\mathbf{e}_{A_t}\|_2} \right\} \subset \bigcup_{k=0}^{\lceil \log_2(\|\mathbf{e}_{A_t}\|_2) \rceil} J_{k,t},$$

where for all interger $1 \leq k \leq \lceil \log_2(\|\mathbf{e}_{A_t}\|_2) \rceil$,

$$J_{k,t} \triangleq \left\{ i \in A_t, \ 2^{1-k}\Lambda_t \geq \frac{2\beta_{i,T}^{1/2}}{N_{i,t-1}^{1/2}} \geq 2^{-k}\Lambda_t \right\}.$$

We can thus upper bound $\Lambda_t^2$ using this decomposition

$$\Lambda_t^2 \leq \left\| \sum_{i \in A_t} \mathbb{I}\left\{ 2\Lambda_t \geq \frac{2\beta_{i,T}^{1/2}}{N_{i,t-1}^{1/2}} \geq \frac{\Lambda_t}{\|\mathbf{e}_{A_t}\|_2} \right\} \frac{2\beta_{i,T}^{1/2} \mathbf{e}_i}{N_{i,t-1}^{1/2}} \right\|_2^2$$

$$\leq \sum_{k=0}^{\lceil \log_2(\|\mathbf{e}_{A_t}\|_2)\rceil} \left\| \sum_{i\in J_{k,t}} \frac{2\beta_{i,T}^{1/2}\mathbf{e}_i}{N_{i,t-1}^{1/2}} \right\|_2^2$$

$$\leq \sum_{k=0}^{\lceil \log_2(\|\mathbf{e}_{A_t}\|_2)\rceil} 2^{2-2k}\Lambda_t^2 \left\| \mathbf{e}_{J_{k,t}} \right\|_2^2.$$

This last inequality implies that there must exist one integer $k_t$ such that $|J_{k_t,t}| = \left\| \mathbf{e}_{J_{k_t,t}} \right\|_2^2 \geq 2^{2k_t-2}(1+\lceil \log_2(\|\mathbf{e}_{A_t}\|_2)\rceil)^{-1}$. We now upper bound $\sum_{t=1}^T \mathbb{I}\{\mathfrak{A}_t\}\Delta_t$, using $|A_t| \leq m$, i.e.,

$$\lceil \log_2(\|\mathbf{e}_{A_t}\|_2)\rceil \leq \lceil \log_2(m)/2\rceil.$$

$$\sum_{t=1}^T \mathbb{I}\{\mathfrak{A}_t\}\Delta_t \leq \sum_{t=1}^T \sum_{k=0}^{\lceil \log_2(m)/2\rceil} \mathbb{I}\{k_t = k,\ \mathfrak{A}_t\}\Delta_t$$

$$\leq \sum_{t=1}^T \sum_{k=0}^{\lceil \log_2(m)/2\rceil} \mathbb{I}\{k_t = k,\ \mathfrak{A}_t\} \sum_{i\in I} \mathbb{I}\{i \in J_{k,t}\}\Delta_t 2^{2-2k}(\lceil \log_2(m)/2\rceil + 1)$$

$$\leq \sum_{t=1}^T \sum_{k=0}^{\lceil \log_2(m)/2\rceil} \sum_{i\in I} \mathbb{I}\left\{ i \in A_t,\ N_{i,t-1}^{1/2} \leq \frac{2^{k+1}\beta_{i,T}^{1/2}}{\Delta_t} \right\}\Delta_t 2^{2-2k}(\lceil \log_2(m)/2\rceil + 1)$$

$$= (\lceil \log_2(m)/2\rceil + 1) \sum_{k=0}^{\lceil \log_2(m)/2\rceil} 2^{2-2k} \sum_{i\in I} \underbrace{\sum_{t=1}^T \mathbb{I}\left\{ i \in A_t,\ N_{i,t-1}^{1/2} \leq \frac{2^{k+1}\beta_{i,T}^{1/2}}{\Delta_t} \right\}\Delta_t}_{(25)_{i,k}}.$$

Applying Proposition 2 gives

$$(25)_{i,k} \leq \frac{\beta_{i,T}2^{\frac{k+1}{1/2}}}{1-1/2}\Delta_{i,\min}^{1-1/1/2}.$$

So we get, using $\lceil \log_2(m)/2\rceil + 1 \leq \log_2(4\sqrt{m})$,

$$\sum_{t=1}^T \mathbb{I}\{\mathfrak{A}_t\}\Delta_t \leq 32\log_2^2(4\sqrt{m}) \sum_{i\in[n]} \beta_{i,T}\Delta_{i,\min}^{-1}.$$

$\square$

The following Proposition 2 is a standard and general result in CMAB, that was first proved in Chen et al. [2013].

**Proposition 2.** *Let* $i \in [n]$ *and* $f_i : \mathbb{R}_+ \to \mathbb{R}_+$ *be a non increasing function, integrable on* $[\Delta_{i,\min}, \Delta_{i,\max}]$. *Then*

$$\sum_{t=1}^T \mathbb{I}\{i \in A_t,\ N_{i,t-1} \leq f_i(\Delta_t)\}\Delta_t \leq f_i(\Delta_{i,\min})\Delta_{i,\min} + \int_{\Delta_{i,\min}}^{\Delta_{i,\max}} f_i(x)\mathrm{d}x.$$

*Proof.* Consider $\Delta_{i,\max} = \Delta_{i,1} \geq \Delta_{i,2} \geq \cdots \geq \Delta_{i,K_i} = \Delta_{i,\min}$ being all possible values for $\Delta_t$ when $i \in A_t$. We define a dummy gap $\Delta_{i,0} = \infty$ and let $f_i(\Delta_{i,0}) = 0$. In (26), we first break the range $(0, f_i(\Delta_t)]$ of the counter $N_{i,t-1}$ into sub intervals:

$$(0, f_i(\Delta_t)] = (f_i(\Delta_{i,0}), f_i(\Delta_{i,1})] \cup \cdots \cup (f_i(\Delta_{i,k_t-1}), f_i(\Delta_{i,k_t})],$$

where $k_t$ is the index such that $\Delta_{i,k_t} = \Delta_t$. This index $k_t$ exists by assumption that the subdivision contains all possible values for $\Delta_t$ when $i \in A_t$. Notice that in (26), we do not explicitly use $k_t$, but instead sum over all $k \in [K_i]$ and filter against the event $\{\Delta_{i,k} \geq \Delta_t\}$, which is equivalent to summing over $k \in [k_t]$.

$$\sum_{t=1}^{T} \mathbb{I}\{i \in A_t,\ N_{i,t-1} \le f_i(\Delta_t)\}\Delta_t$$

$$= \sum_{t=1}^{T} \sum_{k=1}^{K_i} \mathbb{I}\{i \in A_t,\ f_i(\Delta_{i,k-1}) < N_{i,t-1} \le f_i(\Delta_{i,k}), \Delta_{i,k} \ge \Delta_t\}\Delta_t. \qquad (26)$$

Over each event that $N_{i,t-1}$ belongs to the interval $(f_i(\Delta_{i,k-1}), f_i(\Delta_{i,k})]$, we upper bound the suffered gap $\Delta_t$ by $\Delta_{i,k}$.

$$(26) \le \sum_{t=1}^{T} \sum_{k=1}^{K_i} \mathbb{I}\{i \in A_t,\ f_i(\Delta_{i,k-1}) < N_{i,t-1} \le f_i(\Delta_{i,k}), \Delta_{i,k} \ge \Delta_t\}\Delta_{i,k}. \qquad (27)$$

Then, we further upper bound the summation by adding events that $N_{i,t-1}$ belongs to the remaining intervals $(f_i(\Delta_{i,k-1}), f_i(\Delta_{i,k})]$ for $k_t < k \le K_i$, associating them to a suffered gap $\Delta_{i,k}$. This is equivalent to removing the filtering against the event $\{\Delta_{i,k} \ge \Delta_t\}$.

$$(27) \le \sum_{t=1}^{T} \sum_{k=1}^{K_i} \mathbb{I}\{i \in A_t,\ f_i(\Delta_{i,k-1}) < N_{i,t-1} \le f_i(\Delta_{i,k})\}\Delta_{i,k}. \qquad (28)$$

Now, we invert the summation over $t$ and the one over $k$.

$$(28) = \sum_{k=1}^{K_i} \sum_{t=1}^{T} \mathbb{I}\{i \in A_t,\ f_i(\Delta_{i,k-1}) < N_{i,t-1} \le f_i(\Delta_{i,k})\}\Delta_{i,k}. \qquad (29)$$

For each $k \in [K_i]$, the number of times $t \in [T]$ that the counter $N_{i,t-1}$ belongs to $(f_i(\Delta_{i,k-1}), f_i(\Delta_{i,k})]$ can be upper bounded by the number of integers in this interval. This is due to the event $\{i \in A_t\}$, imposing that $N_{i,t-1}$ is incremented, so $N_{i,t-1}$ cannot be worth the same integer for two different times $t$ satisfying $i \in A_t$. We use the fact that for all $x, y \in \mathbb{R}$, $x \le y$, the number of integers in the interval $(x, y]$ is exactly $\lfloor y \rfloor - \lfloor x \rfloor$.

$$(29) \le \sum_{k=1}^{K_i} (\lfloor f_i(\Delta_{i,k}) \rfloor - \lfloor f_i(\Delta_{i,k-1}) \rfloor)\Delta_{i,k}. \qquad (30)$$

We then simply expand the summation, and some terms are cancelled (remember that $f_i(\Delta_{i,0}) = 0$).

$$(30) = \lfloor f_i(\Delta_{i,K_i}) \rfloor \Delta_{i,K_i} + \sum_{k=1}^{K_i-1} \lfloor f_i(\Delta_{i,k}) \rfloor (\Delta_{i,k} - \Delta_{i,k+1}) \qquad (31)$$

We use $\lfloor x \rfloor \le x$ for all $x \in \mathbb{R}$. Finally, we recognize a right Riemann sum, and use the fact that $f_i$ is non increasing to upper bound each $f_i(\Delta_{i,k})(\Delta_{i,k} - \Delta_{i,k+1})$ by $\int_{\Delta_{i,k+1}}^{\Delta_{i,k}} f_i(x)\mathrm{d}x$, for all $k \in [K_i - 1]$.

$$(31) \le f_i(\Delta_{i,K_i})\Delta_{i,K_i} + \sum_{k=1}^{K_i-1} f_i(\Delta_{i,k})(\Delta_{i,k} - \Delta_{i,k+1}) \qquad (32)$$

$$\le f_i(\Delta_{i,K_i})\Delta_{i,K_i} + \int_{\Delta_{i,K_i}}^{\Delta_{i,1}} f_i(x)\mathrm{d}x. \qquad (33)$$

$\square$

There also exist a version for the $\ell_1$-norm error.

**Theorem 5** (Regret bound for $\ell_1$-norm error). *For all $i \in [n]$, let $\beta_{i,T} \in \mathbb{R}_+$. For $t \geq 1$, consider the event*

$$\mathfrak{A}_t \triangleq \left\{ \Delta_t \leq \left\| \sum_{i \in A_t} \frac{\beta_{i,T}^{1/2} \mathbf{e}_i}{N_{i,t-1}^{1/2}} \right\|_1 \right\}.$$

*Then,*

$$\sum_{t=1}^{T} \mathbb{I}\{\mathfrak{A}_t\} \Delta_t \leq \sum_{i \in [n]} \beta_{i,T} 8m \Delta_{i,\min}^{-1}.$$

*Proof.* Let $t \geq 1$. The first step is the reverse amortisation technique, that allows us to modify the upper bound on $\Delta_t$ in such a way that indices $i$ such that $N_{i,t-1}$ is high enough are removed. Assuming that $\mathfrak{A}_t$ holds, we get

$$\Delta_t \leq \sum_{i \in A_t} \mathbb{I}\left\{ \frac{2\beta_{i,T}^{1/2}}{N_{i,t-1}^{1/2}} \geq \frac{\Delta_t}{m} \right\} \frac{2\beta_{i,T}^{1/2}}{N_{i,t-1}^{1/2}}$$

Now, we apply Proposition 3. In summary, we have that $\sum_{t=1}^{T} \mathbb{I}\{\mathfrak{A}_t\} \Delta_t$ is upper bounded by

$$\sum_{i \in [n]} \beta_{i,T} 8m \Delta_{i,\min}^{-1}.$$

$\square$

**Proposition 3.** *Let $i \in [n]$ and $f_i(x) = \beta_{i,T} x^{-1/\alpha_i}$, $\alpha_i \in (0,1]$ and $\beta_{i,T} \geq 0$. Then*

$$\sum_{t=1}^{T} \mathbb{I}\{i \in A_t, \ \delta_t \neq 0, \ N_{i,t-1} \leq f_i(\delta_t)\} f_i^{-1}(N_{i,t-1}) \leq \delta_{i,\min}^{1-1/\alpha_i} \frac{\beta_{i,T}}{1 - \alpha_i} \mathbb{I}\{\alpha_i < 1\}$$

$$+ \mathbb{I}\{\alpha_i = 1\} \beta_{i,T} \left( 1 + \log\left( \frac{\beta_{i,T}}{\delta_{i,\min}} \right) \right).$$

*Proof.* We upper bound $f_i(\delta_t)$ by $f_i(\delta_{i,\min})$ directly in the event, and then simply count the number of integers in $(0, f_i(\delta_{i,\min})]$. For each such integer $s$, the regret suffered is $f_i^{-1}(s)$. We then upper bound the sum by an integral (using the fact that $f_i^{-1}$ is decreasing), to get the final result.

$$\sum_{t=1}^{T} \mathbb{I}\{i \in A_t, \ \delta_t \neq 0, \ N_{i,t-1} \leq f_i(\delta_t)\} f_i^{-1}(N_{i,t-1}) \leq \sum_{t=1}^{T} \mathbb{I}\{i \in A_t, \ N_{i,t-1} \leq f_i(\delta_{i,\min})\} f_i^{-1}(N_{i,t-1})$$

$$\leq \sum_{s=1}^{\lfloor f_i(\delta_{i,\min}) \rfloor} f_i^{-1}(s)$$

$$\leq f_i^{-1}(1) + \int_1^{f_i(\delta_{i,\min})} f_i^{-1}(s) \mathrm{d}s$$

$$= \beta_{i,T}^{\alpha_i} + \int_1^{\beta_{i,T} \delta_{i,\min}^{-1/\alpha_i}} \beta_{i,T}^{\alpha_i} s^{-\alpha_i} \mathrm{d}s$$

$$\leq \mathbb{I}\{\alpha_i < 1\} \delta_{i,\min}^{1-1/\alpha_i} \frac{\beta_{i,T}}{1 - \alpha_i}$$

$$+ \mathbb{I}\{\alpha_i = 1\} \beta_{i,T} \left( 1 + \log\left( \frac{\beta_{i,T}}{\delta_{i,\min}} \right) \right).$$

$\square$

**Proposition 4** (Regret bound for a composed bonus)**.** *Let $K \in \mathbb{N}^*$. For all $t \geq 1$, consider the event*

$$\mathfrak{A}_t \triangleq \left\{ \Delta_t \leq \sum_{k \in [K]} B_{k,t} \right\},$$

*for some $B_{k,t} \geq 0$. Then, the event-filtered regret $\mathbb{E}\left[\sum_{t=1}^T \Delta_t \mathbb{I}\{\mathfrak{A}_t\}\right]$ is upper bounded by*

$$\sum_{k \in [K]} \mathbb{E}\left[ \sum_{t \in [T]} \Delta_t \mathbb{I}\{\Delta_t \leq K B_{k,t}\} \right].$$

*Proof.* From $\mathfrak{A}_t$, there must exists one $k$ such that $\Delta_t \leq K B_{k,t}$. So $1 \leq \sum_{k \in [K]} \mathbb{I}\{\Delta_t \leq K B_{k,t}\}$, i.e., $\Delta_t \leq \sum_{k \in [K]} \Delta_t \mathbb{I}\{\Delta_t \leq K B_{k,t}\}$. $\qquad\square$