[Reviews · NeurIPS 2020]

Review 1

Summary and Contributions: This paper studied the stochastic combinatorial multi-armed bandit (CMAB) problem under two families of distributions: mutually independent outcomes and multivariate sub-Gaussian outcomes. It improved the regret bound of the previous work on Combinatorial Thompson Sampling (CTS) for CMAB under bounded mutually independent outcomes. It proposed a new CTS algorithm with Gaussian priors for a more general sub-Gaussian family and proved a non-trivial upper bound of the algorithm. Experiments demonstrate the effectiveness of the proposed algorithm.

Strengths: It provided a new analysis of CTS under bounded mutually independent outcomes that improve the regret bound from the previous work. For a more general sub-Gaussian family of distribution, it proposed a new algorithm with Gaussian priors and rigorously proved the regret upper bound, which is the main technical contribution. It discussed the insights and the novelty of this regret analysis.

Weaknesses: ---After Rebuttal--- I read the authors' rebuttal and would like to maintain my score of 7. ----- The motivation of considering multivariate sub-Gaussian family of distribution is missing. It is better to explain more or show some concrete applications. The analysis for bounded mutually independent outcomes is still based on the analysis of Wang and Chen [2018], though it modified some lemmas in the proof.

Correctness: This paper is technically sound. Most of the claims are well supported by theoretical analysis.

Clarity: The paper is clearly written and well organized. I suggest double-checking the reference page since some papers are not cited correctly, for example, Agrawal and Goyal [2012], Degenne and Perchet [2016], Wang and Chen [2018] are all published in conferences, not only in arxiv.

Relation to Prior Work: It has detailed comparison with previous works.

Reproducibility: Yes

Additional Feedback:


Review 2

Summary and Contributions: This paper studies the different problem instances of the stochastic combinatorial multi-armed bandits with semi-bandit feedback (CMAB). The considered instances have different dependence structures on arm distributions. The authors propose computationally efficient policies using Combinatorial Thompson Sampling (CTS) policy with Beta priors (CTS-BETA) and Gaussian priors (CTS-GAUSSIAN) for above-mentioned problem instances. They show that the (theoretical and empirical) performance of their policies is better than state-of-art policies.

Strengths: The paper's major contribution is the computationally efficient policy (CTS-GAUSSIAN) for the CMAB problem instance where the joint arm distribution is C-sub-Gaussian, where C is a semi-definite matrix. The authors have shown that the regret upper bound of CTS-GAUSSIAN is the same as state-of-art computationally inefficient ESCB policy. The authors propose CLIP CTS-GAUSSIAN policy for a special case of CMAB problem instance where the joint arm distribution is C-sub-Gaussian with $\lambda \in R^n_+$. The CLIP CTS-GAUSSIAN policy has the same or better regret bounds than the CTS-GAUSSIAN policy. The authors also improve the existing results for the CMAB problem instance where arm distributions are mutually independent.

Weaknesses: CTS-GAUSSIAN policy needs two input parameters: vector D and \beta. It is not clear how to choose vector D if arm distributions are unknown. Also, it is not mentioned that how to select the parameter \beta in the CTS-GAUSSIAN policy.

Correctness: I haven't found any flaws in the paper. I also skimmed the proofs, and it seems sound to me.

Clarity: Overall, the paper is well-written and is easy to follow.

Relation to Prior Work: Authors clearly distinguish between their work from the existing literature.

Reproducibility: Yes

Additional Feedback: Minor comment: 1. Mention the value of \beta used in the experiments. ************************************************************************************* Post Rebuttal: I have read the rebuttal and comments of other reviewers. I will keep my score.


Review 3

Summary and Contributions: The paper considers the stochastic combinatorial multi-armed bandit problem (CMAB), where the learner can play a subset of arms and observes feedback on the arms they played. Motivated by the empirical efficacy of thompson sampling approaches in practice, the paper focuses on developing and analyzing a thompson sampling based approach for CMAB. 1. Assuming the reward distributions of individual arms are independent, the paper improves the regret bound for an existing TS based approach with Beta priors. 2. For the case when the reward distributions of different arms are correlated, the paper proposes a TS based approach with Gaussian priors and proves theoretical bounds.

Strengths: 1. The paper presents a Thompson Sampling (TS) algorithm for the CMAB problem when the rewards from different arms are correlated. Given that the correlated arms is realistic in many CMAB applications and TS is known for its empirical performance, this algorithm would be of larger interest. 2. The paper also improves bounds for existing algorithms in the case of arms having independent rewards.

Weaknesses: The technical contributions in this paper are marginal for a theoretical work. 1. The improvement in bounds for the TS-Beta algorithm are marginal and the techniques used doesn't apply to other problems. 2. Similarly, the algorithm presented for the case of correlated rewards is a natural generalization of well known algorithm for the stochastic MAB problem. I believe the second algorithm could be improved if we can sample from a multi-variate gaussian prior instead of the current choice of independent gaussian priors. 3. I think the paper can also improve the writing, particularly articulating the implications of correlated priors vs independent priors.

Correctness: Yes

Clarity: There's room for improvement in the writing. 1. Clearly articulating challenges involving in generalizing TS based approaches to correlated rewards setting and how the paper resolved them. 2. Discussing the implication of different priors (correlated vs independent) on performance (even if it's empirical).

Relation to Prior Work: Largely, the paper does a good job, but some of the issues with writing also apply to discussing prior work.

Reproducibility: Yes

Additional Feedback: Missing reference: Thompson Sampling for the MNL Bandit. This paper discusses some interesting algorithmic choices for a very stylized CMAB problem. In particular, this paper shows that there's value in sampling parameters using a common posterior distribution instead of the regular independent sampling. Post rebuttal: I've read author's response, other reviewers comments and have taken these into account to maintain my original score. It would be good if the paper can adapt some of these ideas and compare it with the performance of Algorithm-2.


Review 4

Summary and Contributions: This paper studies stochastic combinatorial multi-armed bandit with semi-bandit feedback (CMAB). This paper lists three kinds of settings and mainly focuses on setting (i) and (iii). This paper first improves the result of Wang and Chen [2018] by providing the regret upper bound for CTS-BETA in the setting (i) with bounded outcomes and it also holds for nonlinear reward functions. Moreover, the authors propose a new policy called CTS-GAUSSIAN and give a regret bound reducing to O(log^2(m) n log(T)/\Delta) for a diagonal sub-Gaussian matrix. When the reward is linear, this paper proposes CLIP CTS-GAUSSIAN and give an improved regret bound. Experiments on applications show the superiority of the policies.

Strengths: - The problem is well defined and has some prior results in the literature. This paper not only improves the prior results for setting (i) but also fill the blank for setting (iii). - The new assumptions introduced in this paper is natural and acceptable. - Besides the rigorously proved better theoretical guarantee, this paper also conducts experiment to compare with the baselines in two applications, which shows that the proposed policy has the better performance and less computational time.

Weaknesses: - Since there still exist some gaps between the upper bound and lower bound, do the proposed upper bounds tight?

Correctness: I didn't go through the proofs in appendix.

Clarity: Yes.

Relation to Prior Work: Yes. This paper clearly compares with the prior work, ie. Wang and Chen [2018].

Reproducibility: Yes

Additional Feedback:

[Author Response · NeurIPS 2020]



Figure 1: Comparison with correlated prior sampling for the $K_{4,4}$ matching problem, with $c = 0, 0.5, 1$.

We want to thank the reviewers for their useful and positive feedbacks. We answer their comments/questions
in the following paragraphs, that will be incorporated in the paper.

**Correlated vs independent prior**    We briefly discussed the use of a correlated prior in future work and
also in footnote 3, mentioning that the policy would perform better than using an independent prior. We ran
additional empirical comparisons to assess this, plotting the results in Figure 1. As expected, the correlated
prior policy is better (when outcomes are correlated). This motivates the theoretical study of such policy for
future work. However, some new challenges are raised, as we see in the following paragraph.

**Importance of using a factorized prior in our analysis**    A factorized prior allows us to bound the
filtered regret against the event $\mathfrak{S}_t(Z) \wedge \mathfrak{T}_t(Z)$ (see p. 13, beginning of step 4). More precisely, we count the
number of rounds needed for $\mathfrak{S}_t(Z) \wedge \neg\mathfrak{T}_t(Z)$ to occur for the $q$-th time. Under such event, the arms of $Z$
are all observed, i.e., the corresponding priors are updated and counters are thus lower bounded by $q$. The
importance of having a factorize prior resides in bounding the expected number of rounds for $\mathfrak{S}_t(Z) \wedge \neg\mathfrak{T}_t(Z)$
to occur. Indeed, this is (essentially) equal to $\mathbb{E}[1/\mathbb{P}[\neg\mathfrak{T}_t(Z)|\mathfrak{S}_t(Z)]]$. Since $\mathfrak{S}_t(Z)$ and $\mathfrak{T}_t(Z)$ are independent,
this is further equal to $\mathbb{E}[1/\mathbb{P}[\neg\mathfrak{T}_t(Z)]]$, allowing us to conduct our analysis by showing that this last quantity
is exponentially decreasing in $q$ (so summable over $q$). To the best of our knowledge, it is unknown how to
get such a bound when $\mathfrak{S}_t(Z)$ and $\mathfrak{T}_t(Z)$ are not independent.

**Technical contributions for cts-beta**    Although the gain in the upper bound for the beta prior might be
considered as marginal, we want to stress on the asymptotic (quasi) optimality of the new bound (considering
that the $\log^2(m)$ factor is negligible compared to $n$). We also fixed some technical issues in the proof of the
previous bound; as a consequence, we believe that, although our work on CTS-BETA might appear somewhat
incremental over Wang and Chen [2018], it brings essential clarifications to the literature.

**Technical contributions for cts-gaussian**    Although the CTS-GAUSSIAN policies[1] are natural and essen-
tially not new, our contribution is in their analyses, that are non-incremental. In particular, the stochastic
dominance method is completely new, and allows us to convert correlated outcomes into independent Gaussian
ones. Independence is crucial to be able to factorize the expectation $\mathbb{E}[1/\mathbb{P}[\neg\mathfrak{T}_t(Z)]]$. To the best of our
knowledge, asymptotic quasi optimal analysis only exists for a restrictive class of action spaces (matroid) or
outcomes distributions (independent). Dealing with both a general action space and outcome distribution is
challenging, and represents our main contribution.

**Motivation for sub-Gaussian outcomes**    In the same way as boundedness generalizes to sub-Gaussianity
in 1d, we have that if $\mathbf{X}$ is a.s. in a compact $\mathcal{K}$, it is $\mathbf{C}$-sub-Gaussian, with $\mathbf{C}$ built from the John's ellipsoid
of $\mathcal{K}$. In this case, $D_i$ is computed with a linear maximization over $\mathcal{A}$ (see footnote 4). In particular,
$\mathcal{K} = B_{\ell_\infty}(0, 1)$ gives $D_i = m$, and $\mathcal{K} = B_{\ell_2}(0, 1)$ gives $D_i = 1$. We can also use other structures on the
outcomes to have $D_i$, such as negative dependence (as in our shortest path experiments).

**Comparison to previous work**    We will add a comparison to Thompson Sampling for the MNL Bandit
in the revised version, mentioning that the use of a common posterior distribution boosts the probability
that the samples are all optimistic, and can thus greatly improve the constant term in our bound. However,
as we saw, obtaining a quasi optimal gap dependent bound for such correlated sampling is an open question.

## Footnotes

[1] $\beta > 1$ is an artefact of the analysis and can in practice be taken equal to 1 (as we did in our experiments).


[Meta-Review · NeurIPS 2020]

The paper has been received three positive review and one lukewarm/slightly negative review. The lukewarm review was very informative and the concerns raised were mostly admitted by authors. The main concerns were that the contributions beyond the previous work is incremental in the sense that it only fills some gap in the literature and does not make very impactful contribution. I agree with this criticism even after reading author's response. The authors argue that the use of stochastic dominance technique is novel, however, the use of stochastic optimism in previous bandit papers (e.g., several papers by Osband and Van Roy) involves similar arguments. Having said that, I do think that the paper is well written, the results are correct, and the results will be of interest to the community working on this problem. Overall, the paper meets the bar for publication.